# Using the Maximum Entropy Production approach to integrate energy budget modeling in a hydrological model

Audrey Maheu[1], Islem Hajji[2], François Anctil[2], Daniel F. Nadeau[2], René Therrien[3]

[1]Département des sciences naturelles, Université du Québec en Outaouais, Ripon, J0V 1V0, Canada
[2]Département de génie civil et de génie des eaux, Université Laval, Québec, G1V 0A6, Canada
[3]Département de géologie et de génie géologique, Université Laval, Québec, G1V 0A6, Canada

*Correspondence to*: Audrey Maheu (audrey.maheu@uqo.ca)

**Abstract.** Total terrestrial evaporation is a key process to understand the hydrological impacts of climate change given that warmer surface temperatures translate into an increase in the atmospheric evaporative demand. To simulate this flux, many
hydrological models rely on the concept of potential evaporation (PET) although large differences have been observed in the response of PET models to climate change. The Maximum Entropy Production (MEP) model of land surface fluxes offers an alternative approach to simulate terrestrial evaporation in a simple way while fulfilling the physical constraint of energy budget closure and providing a distinct estimation of evaporation and transpiration. The objective of this work is to use the MEP model to integrate energy budget modeling within a hydrological model. We coupled the MEP model with
HydroGeoSphere, an integrated surface and subsurface hydrologic model. As a proof-of-concept, we performed one-dimensional soil column simulations at three sites of the AmeriFlux network. The coupled HGS-MEP model produced realistic simulations of soil water content (RMSE between 0.03 and 0.05 $m^3$ $m^{-3}$, NSE between 0.30 and 0.92) and terrestrial evaporation (RMSE between 0.31 and 0.71 mm $day^{-1}$, NSE between 0.65 and 0.88) under semiarid, Mediterranean and temperate climates. At the daily time scale, HGS-MEP outperformed the standalone HGS model where total terrestrial
evaporation is derived from potential evaporation which we computed using the Penman-Monteith equation, although both models had comparable performance at the half-hourly time scale. This research demonstrated the potential of the MEP model to improve the simulation of total terrestrial evaporation in hydrological models, including for hydrological projections under climate change.

## 1 Introduction

Driven by climate change, warmer surface temperatures are expected to increase the atmospheric evaporative demand (Breshears et al., 2013; Ficklin and Novick, 2017). As such, total terrestrial evaporation (herein after terrestrial evaporation), or "evapotranspiration", is a key process to assess the impacts of climate change on stream discharge. To simulate this flux, many hydrological models rely on the concept of potential evaporation (PET), which is defined as the maximum evaporation that can occur under ambient meteorological conditions with an unlimited water supply. Two types of PET models are
generally used in hydrological models: temperature-based models where PET is estimated via air temperature (Thornthwaite, 1948; Hamon, 1963) and physically based models where PET estimation is based on components of the surface energy budget (Monteith, 1965; Priestley and Taylor, 1972). Various studies have compared PET models to assess their suitable range of application (Xu and Singh, 2001). While the performance of PET models varies across regions, the choice of a PET model appears to have minimal influence on stream discharge simulation under contemporary climate conditions
(Andréassian et al., 2004; Oudin et al., 2005; Isabelle et al., 2018). However, when future climate projections are incorporated in PET models, the predicted PET varies significantly from model to model (McKenney and Rosenberg, 1993; Kingston et al., 2009; Donohue et al., 2010; Lofgren et al., 2011; McAfee, 2013; Hosseinzadehtalaei et al., 2016). These differences often translate into large uncertainty in stream discharge projections (Kay and Davies, 2008; Bae et al., 2011; Milly and Dunne, 2011; Prudhomme and Williamson, 2013; Seiller and Anctil, 2016), although some studies have
demonstrated opposite results with a relatively low sensitivity of discharge projection to PET model selection (Thompson et al., 2014; Koedyk and Kingston, 2016).

Among PET models, those that are temperature-based tend to mainly reflect changes in mean air temperature, even though temperature is not necessarily the strongest control on PET (Donohue et al., 2010; Shaw and Riha, 2011). For example, the oversensitivity of temperature-based models to surface warming has been linked to an exaggerated assessment of drought severity under climate change (Hoerling et al., 2012). As a result, recent studies have recommended the use of physically based PET models, such as the Penman model, to simulate terrestrial evaporation in climate change assessments (Hobbins et al., 2008; Sheffield et al., 2012; McAfee, 2013). However, additional data inputs (e.g., radiation, humidity and wind speed) are required to implement such models and, as opposed to the high confidence put in temperature estimates from global climate models (McMahon et al., 2015), these variables usually suffer from a lower reliability. For example, large differences persist between observations, reanalysis products and global climate model projections of wind speed (McVicar et al., 2008; Pryor et al., 2009). Overall, a dilemma emerges for the simulation of PET under climate change, which involves choosing between a reliable physically based model with greater input data uncertainty versus an overly simplified temperature-based model with reliable input data (Ekström et al., 2007; Kay and Davies, 2008; Kingston et al., 2009).

Given these current limitations with PET models, alternative approaches are needed for the simulation of terrestrial evaporation in climate change assessments. Energy budget modeling offers a physically based approach to simulate terrestrial evaporation (along with other components of the surface energy budget), with the added benefit of conservation of energy that ensures a balance between incoming and outgoing energy. Land surface models provide a means to simulate both water and energy budgets and they have been used for hydrological modeling, either by i) coupling hydrological and land surface models (Pietroniro et al., 2001; Maxwell and Miller, 2005; Kunstmann et al., 2008; Zabel and Mauser, 2013) or by ii) coupling a land surface model to a routing scheme (Gaborit et al., 2017). In land surface models, terrestrial evaporation is generally estimated with a bulk aerodynamic approach (Noilhan and Planton, 1989) that relies on semi-empirical equations and requires information on vertical gradients of air temperature and humidity that can introduce substantial errors. The use of land surface models can yield small gains in hydrological modeling performance (Livneh et al., 2011; Shi et al., 2014), but they remain computationally heavy.

Optimality principles, which are derived from the idea that nature organizes itself to ensure optimal functioning, provide an alternative avenue to model terrestrial evaporation. Maximum entropy production (MEP) is one of the optimality principles put forward and, although it has yet to become an established principle, it offers a promising method to improve hydrological modeling (Ehret et al., 2014; Westhoff and Zehe, 2013). The MEP principle has been applied in two ways in hydrology. In the first approach, MEP is defined as a physical principle, which hypothesizes that an open thermodynamic system far from equilibrium can achieve a steady state at which, through self-organization, entropy is produced at the maximum possible rate. Such a dynamic equilibrium can be achieved because, as demonstrated by Paltridge (1975), a gradient drives a flux that in turn, depletes the initial gradient and further reinforces it. Using this approach, MEP has been used to constrain parameters of hydrological models (Kleidon and Schymanski, 2008; Porada et al., 2011; Westhoff and Zehe, 2013). In the second approach, MEP is defined as a statistical principle and constitutes the most probable state of an open system (Dewar, 2005; 2009). Using the MaxEnt statistical inference algorithm (Jaynes, 1957), the state of MEP can be predicted by maximizing Shannon information entropy while considering constraints imposed by the available information. Both concepts of entropy are linked: the maximisation of Shannon information entropy can be used to assess the probability of a given state for any kind of system and as such, thermodynamic entropy can be considered a special case of Shannon information theory. In hydrology, Wang and Bras (2009, 2011) have used the Shannon information entropy approach to develop the MEP model of land surface fluxes, which is the topic of the present article.

The MEP model of land surface fluxes (Wang and Bras, 2009, 2011) offers a simple approach for energy budget modeling that ensures the closure of the surface energy balance. In the MEP model developed by Wang and Bras, only three input variables, net radiation, surface temperature and surface specific humidity, are required to model surface heat fluxes, with a different definition of the surface when assessing evaporation (surface = soil) and transpiration (surface = leaf). As it will be presented below, in practice, the following six input variables are needed to operate MEP continuously under all climate conditions: net radiation, soil surface temperature and specific humidity, leaf surface temperature and specific humidity and a vegetation index. Overall, the MEP model eliminates the need for wind speed and surface roughness data (i.e. input data to

the Penman model) as well as eliminating the need for data on vertical gradients (i.e. temperature and humidity, input data to the bulk aerodynamic model often implemented in land surface models). Moreover, the predicted surface heat fluxes are constrained by the available energy (i.e. net radiation), which avoids the issue of oversensitivity to temperature and suggests that this approach should be robust for climate change assessments. Recent studies also reported improved performance of the

MEP approach for the simulation of terrestrial evaporation when compared against the Penman-Monteith model (Hajji et al., 2018) or a modified Penman-Monteith approach driven by remote sensing and reanalysis data (Xu et al., 2019). The MEP model also had a performance comparable to a complex land surface model (Canadian Land Surface Scheme, CLASS) at snow-free sites at low latitudes (Alves et al., 2019). Overall, the MEP model offers an attractive approach to improve the simulation of terrestrial evaporation in hydrological modeling and to increase the robustness of streamflow projections under

climate change. The objective of this study is thus to couple the MEP model of land surface fluxes with a hydrological model and perform proof-of-concept simulations at AmeriFlux sites spanning a range of climatic and vegetation conditions.

### 2 Study area

We selected three sites of the AmeriFlux network (Baldocchi et al., 2001) where the eddy covariance method is used to measure vertical water and energy fluxes (Table 1, Figure 1). We selected sites using the following criteria: i) measurements

of volumetric soil water content available at different depths and for an extended period; ii) absence of snow cover and iii) diversity in climates (semiarid, Mediterranean, temperate) and types of vegetation (grassland, woody savanna, deciduous broadleaf forest).

**Table 1. Description of study sites**

|  | US-Wkg | US-Ton | US-WBW |
|---|---|---|---|
| climate | semiarid | Mediterranean | temperate |
| Koppen climate class | Bsk: steppe, warm winter | Csa: Mediterranean, mild with dry, hot summer | Cfa: humid subtropical, mild with no dry season, hot summer |
| IGPB vegetation class | grassland | woody savanna | deciduous broadleaf forest |
| mean annual precipitation (mm) | 407 | 559 | 1372 |
| mean annual temperature (°C) | 15.6 | 15.8 | 13.7 |
| data source | Scott (2016) | Baldocchi (2016) | Meyers (2016) |

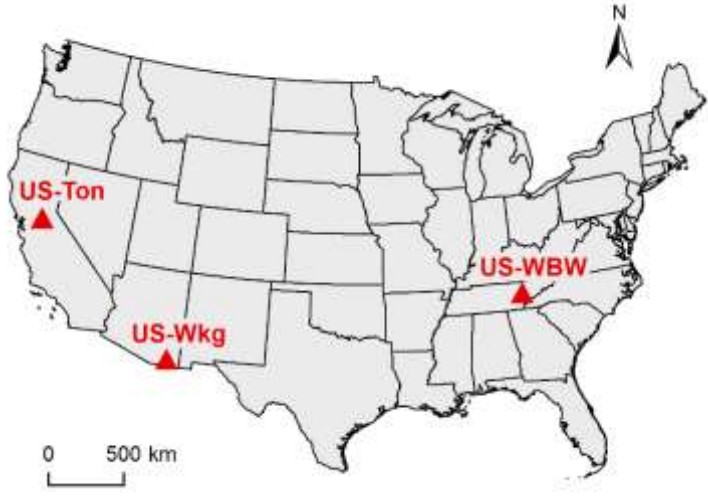

**Figure 1. Location of study sites across the conterminous United States**

The first site, US-Wkg, is characterized by a semiarid climate where 60% of precipitation is concentrated in July, August, and September. In 2006, a sharp transition occurred in the vegetation and the native grassland species (assemblage of grama grasses, genus *Bouteloua*) have been supplanted by Lehman lovegrass (*Eragrostis lehmanniana*), an exotic species spreading throughout southwestern United States (Moran et al., 2009; Scott et al., 2010). The second site, US-Ton, is characterized by a Mediterranean climate with dry summers and most precipitation occurring from October to May. The woody savanna is made up of two layers of vegetation, each reaching peak activity at different times of the year (Baldocchi et al., 2004). Trees, mainly blue oaks (*Quercus douglasii*), are dormant during winter months, reach peak activity in the spring and then carefully regulate their water use during the dry summer period. In contrast, the understory vegetation composed of grasses and forbs is mainly active during the rainy winter period when water is plentiful. Finally, the US-WBW site is characterized by a temperate climate with precipitation relatively evenly distributed throughout the year. The mixed deciduous forest is dominated by oak and hickory and vegetation is active from the spring to early autumn (April to October), with peak activity during the summer (Wilson and Baldocchi, 2000).

## 3 Methodology

### 3.1 Maximum entropy production (MEP) model

The MEP model of land surface fluxes uses the optimality principle of maximum entropy production (Dewar, 2005) as an inference tool for nonequilibrium thermodynamic systems. In the MEP model, entropy refers to Shannon entropy, the expected value of information (Shannon, 1948), which is not related to thermodynamic entropy expressed as the ratio of flux to temperature. Indeed, the MEP model is derived from the principle of maximum entropy (MaxEnt) developed by Jaynes (1957) as a general inference tool to assign probability distributions in statistical mechanics. Wang and Bras (2009) used this statistical approach to develop the MEP model of land surface fluxes. Within the MEP model, land surface fluxes are each expressed in terms of a dissipation or entropy function and, under the constraint of conservation of energy, a unique extremum solution is found.

Dissipation functions have been postulated for land surface fluxes over a dry soil (Wang and Bras, 2009) and over wet soil and vegetation surfaces (Wang and Bras, 2011). For non-vegetated surfaces, bare soil evaporation ($E_s$) is estimated by solving equations 1 and 2 with the constraint of energy conservation at the surface ($R_n = G + E_s + H$) (Wang and Bras, 2011):

$$E_s = B(\sigma)H \tag{1}$$

$$G = \frac{B(\sigma)}{\sigma} \frac{I_s}{I_0} H |H|^{-\frac{1}{6}} \tag{2}$$

where $R_n$, $E_s$, $H$ and $G$ are the net radiation, latent, sensible and ground heat fluxes at the soil surface (W m$^{-2}$; positive values indicate a heat flux away from surface), $B(\sigma)$ is the inverse of the Bowen ratio (see equation 7), $\sigma$ is a dimensionless parameter (see equation 8). $I_s$ is the soil thermal inertia (J m$^{-2}$ K$^{-1}$ s$^{-1/2}$), calculated with the empirically derived equation from Huang and Wang (2016):

$$I_s = \sqrt{I_{ds}^2 + \theta I_w^2} \tag{3}$$

where $I_{ds}$ is the dry soil thermal inertia (J m$^{-2}$ K$^{-1}$ s$^{-1/2}$), $\theta$ is the volumetric soil moisture (m$^3$ m$^{-3}$) and $I_w$ is the thermal inertia of liquid water (1557 J m$^{-2}$ K$^{-1}$ s$^{-1/2}$). $I_0$ is the "apparent thermal inertia of air" (J m$^{-2}$ K$^{-1}$ s$^{-1/2}$) and was calculated as:

$$I_0 = \rho_a c_p \sqrt{C_1 kz} \left( C_2 \frac{kzg}{\rho_a c_p T_0} \right)^{1/6} \tag{4}$$

where $\rho_a$ is the air density (1.22 kg m$^{-3}$), $c_p$ is specific heat of air under constant pressure (1004 J kg$^{-1}$ K$^{-1}$), $C_1$ and $C_2$ are two constants in the empirical functions representing the effects of stability on the mean profiles of wind speed and

temperature within the surface layer (Wang and Bras, 2009), $k$ is the von Kármán constant (0.4), $z$ is the height above ground based on vegetation conditions (m), $g$ is the gravitational acceleration (9.81 m s$^{-2}$) and $T_0$ is a reference temperature (300 K).

For vegetated surfaces, the MEP model of transpiration considers the energy balance at the vegetation surface. As such, the term $G$ corresponds to the canopy heat flux (not to be confused with the ground heat flux $G$ in the MEP model for non-vegetated surfaces, equations 1-2) and is considered negligible compared to the sensible and latent heat fluxes (Wang and Bras, 2011). Accordingly, the energy balance is simplified and transpiration ($E_t$) is estimated by solving equations 5 and 6 under the constraint of energy conservation at the surface ($R_n = E_t + H$):

$$E_t = \frac{R_n}{1+B^{-1}(\sigma)} \tag{5}$$

$$H = \frac{R_n}{1+B(\sigma)} \tag{6}$$

The same definition of $B(\sigma)$ and $\sigma$ applies for non-vegetated (equations 1-2) and vegetated (equations 5-6) surfaces and the Bowen reciprocal ratio is calculated as:

$$B(\sigma) = 6\left(\sqrt{1+\frac{11}{36}\sigma} - 1\right) \tag{7}$$

where $\sigma$ corresponds to a dimensionless parameter that characterizes the phase change at the evaporating or transpiring surface:

$$\sigma = \frac{\sqrt{\alpha}\lambda^2}{c_p R_v}\frac{q_s}{T_s^2} \tag{8}$$

where $\alpha$ is the ratio of the eddy diffusivities for water vapor and heat (assumed to be unity, (Wang et al., 2014), $\lambda$ is the latent heat of vaporization of liquid water (J kg$^{-1}$), $C_p$ is the specific heat of air under constant pressure (J kg$^{-1}$ K$^{-1}$), $R_v$ is the gas constant of water vapor (461 J kg$^{-1}$ K$^{-1}$), $q_s$ is the surface specific humidity (kg kg$^{-1}$), and $T_s$ is the surface temperature (K).

In the MEP model of evaporation (MEP-E$_s$, equations 1-2), the surface specific humidity ($q_s$) corresponds to the specific humidity at the soil surface ($q_{ss}$) and the surface temperature ($T_s$) corresponds to skin temperature at the soil surface ($T_{ss}$). Equation 8 can be modified accordingly:

$$\sigma = \frac{\lambda^2}{c_p R_v}\frac{q_{ss}}{T_{ss}^2} \tag{9}$$

The specific humidity at the soil surface can be computed as (Huang and Wang, 2016):

$$q_{ss} = \left(\frac{\theta}{\theta_s}\right)^\beta q_{sat} \tag{10}$$

where $q_{sat}$ is the specific humidity at saturation (kg kg$^{-1}$) at $T_{ss}$ which is calculated using the Clausius-Clapeyron equation, $\theta$ is the soil water content (m$^3$ m$^{-3}$), $\theta_s$ is the soil porosity (m$^3$ m$^{-3}$) and $\beta$ is an empirical parameter that was set to $\beta = 2$ based on Huang and Wang (2016).

In the MEP model of transpiration (MEP-E$_t$, equations 5-6), the surface specific humidity ($q_s$) corresponds to the specific humidity at the leaf surface ($q_{ls}$) and the surface temperature ($T_s$) corresponds to the leaf temperature ($T_{ls}$). A water stress factor ($\eta_s$) that varies between 0 and 1 is added to the original formulation of equation 8 presented by Wang and Bras (2011) to describe the reduction in plant transpiration under soil water stress (Hajji et al., 2018). For MEP-E$_t$, equation 8 thus translates into:

$$\sigma = \eta_s \frac{\lambda^2}{c_p R_v}\frac{q_{ls}}{T_{ls}^2} \tag{11}$$

Various parameterizations of $\eta_s$ exist in the literature based on soil water potential (Verhoef and Egea, 2014; Ferguson et al., 2016), volumetric soil water content (Feddes et al., 1978; Porporato et al., 2001) or leaf water potential (Tuzet et al., 2003). We used the Wang and Leuning (1998) parameterization of $\eta_s$ based on soil water content that Hajji et al. (2018) successfully used to apply the MEP model of transpiration under water-limiting conditions:

$$\eta_s = \min\left[1, \frac{10\,(\theta_{root} - \theta_{wp})}{3\,(\theta_{fc} - \theta_{wp})}\right] \tag{12}$$

where $\theta_{root}$ is the weighted average soil water content over the root zone, with weights set according to the vertical root distribution (m$^3$ m$^{-3}$), $\theta_{wp}$ is the soil water content at wilting point (m$^3$ m$^{-3}$) and $\theta_{fc}$ is the soil water content at field capacity (m$^3$ m$^{-3}$).

The MEP models have been formulated for the two limiting cases of bare soil evaporation and transpiration from a fully vegetated surface. In order to continuously simulate terrestrial evaporation, Hajji et al. (2018) proposed a method to combine the MEP models of evaporation and transpiration using a vegetation index ($f_{veg}$) that corresponds to the fraction of the soil covered by vegetation. Assuming that evaporation from intercepted rainfall is negligible, as is the case in this study, the MEP model of terrestrial evaporation (MEP-E) is then defined as:

$$E = \left(1 - f_{veg}\right)E_s + f_{veg}E_t \tag{13}$$

The vegetation index ($f_{veg}$) can be derived from the Normalized Difference Vegetation Index (NDVI) (Gutman and Ignatov, 1998):

$$f_{veg} = \frac{NDVI_t - NDVI_{\min}}{NDVI_{\max} - NDVI_{\min}} \tag{14}$$

where $NDVI_t$ is the NDVI on day $t$, $NDVI_{\min}$ is the NDVI signal from bare soil and $NDVI_{\max}$ is the NDVI signal from a full vegetation cover.

## 3.2 Hydrological model

HydroGeoSphere (HGS) is an integrated surface and subsurface hydrologic model that has been used to simulate soil water content at various AmeriFlux sites (Maheu et al., 2018) as well as in a forested headwater catchment (Koch et al., 2016). Maxwell et al. (2014) performed a formal verification of seven integrated surface and subsurface models and HGS showed good agreement with other models when simulating soil moisture in an idealized test case. HGS is a control-volume finite-element model that simultaneously solves the 2-D diffusion-wave approximation of the Saint-Venant equations and the 3-D form of the Richards equation (Aquanty, 2013). The surface and subsurface are coupled via a first-order exchange coefficient. The commonly used van Genuchten (1980) model is incorporated in HGS to describe the water retention curve. HGS has an adaptive time-stepping procedure where the time step length is controlled by the maximum change allowed in a state variable during any time step. In the present study, we allowed for a maximum change of 0.05 in soil saturation. In the current version of HGS, transpiration and evaporation are simulated as a function of potential evaporation (Kristensen and Jensen, 1975):

$$E_t(z) = f_1(LAI)\,f_2(\theta)\left(E_p - E_c\right)r(z) \tag{15}$$

where $E_t(z)$ is the transpiration rate (m s$^{-1}$) at depth $z$, $E_p$ is the potential evaporation rate (m s$^{-1}$), $E_c$ is the wet canopy evaporation rate (m s$^{-1}$). $f_1(LAI)$ is a function of the leaf area index (LAI) representing changes in vegetation over time (dimensionless):

$$f_1(LAI) = max\{0, min[1, C_2 + C_1 LAI]\} \tag{16}$$

where $C_1$ and $C_2$ are dimensionless fitting parameters. $f_2(\theta)$ is a function describing the water stress on vegetation (dimensionless) which is equivalent to $\eta_s$ (equation 12). $r(z)$ is the root distribution function which follows a cubic decay distribution between the surface and the maximum root depth. Soil evaporation is described as:

$$E_s(z) = \alpha^* [1 - f_1(LAI)] (E_p - E_c) e(z) \tag{17}$$

where $E_s(z)$ is the soil evaporation rate (m s$^{-1}$) at depth $z$, $e(z)$ is the evaporation depth function which follows a cubic decay distribution between the surface and the maximum evaporation depth. $\alpha^*$ is a soil wetness factor (dimensionless):

$$\alpha^* = \begin{cases} \frac{\theta - \theta_{e2}}{\theta_{e1} - \theta_{e2}} & \text{for } \theta_{e2} \le \theta \le \theta_{e1} \\ 1 & \text{for } \theta > \theta_{e1} \\ 0 & \text{for } \theta < \theta_{e2} \end{cases} \tag{18}$$

where $\theta_{e1}$ is the soil water content above which full evaporation occurs and $\theta_{e2}$ is the soil water content below which evaporation is zero.

### 3.3 Coupling the HGS and MEP-E models

In the HGS model, the following modified form of Richards equation is used to describe the temporal evolution of soil moisture:

$$-\nabla \cdot \mathbf{q} + \sum \Gamma_{ex} \pm Q = \frac{\partial}{\partial t} (\theta_s S_w) \tag{19}$$

where $\mathbf{q}$ is the specific volumetric (Darcy) flux (m$^3$ s$^{-1}$), $\Gamma_{ex}$ is the source/sink term for exchange fluxes with other domains (i.e. surface) (m$^3$ s$^{-1}$), $Q$ is the source/sink term for the porous medium such as pumping or injection (m$^3$ s$^{-1}$) and $S_w$ is the subsurface water saturation (m$^3$ m$^{-3}$). The MEP-E$_s$ model requires soil moisture information to compute three parameters: the soil thermal inertia ($I_s$, equation 3) and the surface specific humidity of the soil ($q_{ss}$, equation 9) and leaf ($q_{ls}$, equation 11)
surfaces. At the beginning of a given time step, HGS supplies the soil moisture information from the previous time step to the MEP-E model which then computes the evaporation and transpiration rates (Figure 2). The transpiration and evaporation rates computed with the MEP-E model are then transferred to HGS and taken as sinks for the porous medium ($\Gamma_{ex}$ in equation 19). HGS removes water from the soil reservoir based on the root and evaporation depth profiles, which both follow a cubic decay distribution between the surface and the maximum root or evaporation depth. At the end of the given time step, HGS
calculates the saturation throughout the soil column. Given that the MEP-E$_s$ model was not very sensitive to changes in soil thermal inertia, this parameter was set as a constant equal to the dry soil thermal inertia (as detailed in the next section) and was not involved in the coupling procedure.

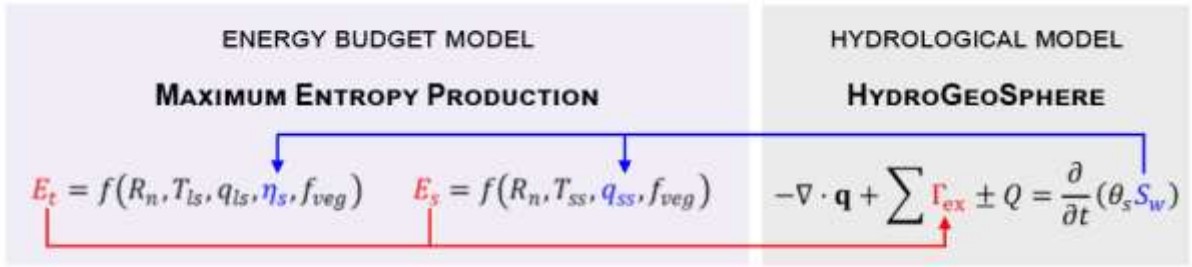

**Figure 2. Coupling of the HydroGeoSphere (HGS) hydrological model and the Maximum Entropy Production (MEP) model of land surface fluxes. HGS supplies the MEP model with soil water saturations ($S_w$, in blue) which are used by the MEP model to compute the transpiration ($E_t$) and evaporation ($E_s$) rates. These rates are then passed on to HGS to compute the sink terms with the surface ($\Gamma_{ex}$, in red) and HGS removes water from the soil reservoir based on the root and evaporation depth profiles.**

### 3.4 Model implementation

### 3.4.1 HGS-MEP

The coupled HGS-MEP model was not calibrated in the present study. The soil and vegetation properties were therefore not obtained from calibration but were rather defined from pedo-transfer functions, observations at AmeriFlux sites or values taken from the literature. We derived soil hydraulic characteristics (Table 2) using soil texture information as an input to the Rosetta model implemented in Hydrus (Šimůnek et al., 2013) to derive the soil hydraulic characteristics. Soil texture information was either taken from the AmeriFlux database (US-Ton) or from the literature (US-Wkg in Nearing et al., 2005 and US-WBW in Miller et al., 2007). At US-Wkg and US-Ton, the observed minimum soil water content was smaller than residual saturation computed by the Rosetta model and, as such, we set the residual saturation in HGS to the observed minimum soil water content. At each site, we defined soil porosity as the observed maximum soil water content closest to the surface and we used the water retention curve to compute the volumetric water content corresponding to the wilting point (–1.5 MPa) and field capacity (–0.033 MPa). We defined the maximum root depth using site-specific information at US-Ton (Ichii et al., 2009) and US-WBW (Wilson et al., 2001). Given that similar information was unavailable at US-Wkg, we set the maximum root depth based on the root depth reported for Lehman lovegrass in the literature (Gibbens and Lenz, 2001), the main grass species at the site. Given only little information is available on soil and root properties, parameter allocation for these properties is often an important source of uncertainty in hydrological models. However, previous modeling work at the study sites has shown soil moisture modeling to be relatively robust to variations in saturated hydraulic conductivity and vertical root distribution (Maheu et al., 2018).

**Table 2. Soil and vegetation properties at each study site**

|  | US-Wkg | US-Ton | US-WBW |
|---|---|---|---|
| % sand – silt – clay [a] | 55 – 20 – 25 | 48 – 42 – 10 | 28 – 60 – 12 |
| saturated hydraulic conductivity (x$10^{-6}$ m s$^{-1}$) | 2.813 | 2.929 | 3.615 |
| residual saturation (m$^3$ m$^{-3}$) | 0.01 | 0.02 | 0.05 |
| wilting point (m$^3$ m$^{-3}$) | 0.05 | 0.07 | 0.07 |
| field capacity (m$^3$ m$^{-3}$) | 0.21 | 0.29 | 0.26 |
| porosity (m$^3$ m$^{-3}$) | 0.44 | 0.56 | 0.38 |
| root depth (m) | 1.2 | 0.7 | 0.6 |

[a] US-Wkg: Nearing et al. (2005); US-Ton: BADM from the AmeriFlux database; US-WBW: Miller et al. (2007)

Table 3 provides a summary of the implementation of the MEP model of terrestrial evaporation in HGS. In the MEP model of evaporation, skin temperature measurements would ideally be used to set the soil surface temperature ($T_{ss}$) but, since those measurements were not available, we instead used soil temperature measurements nearest to the surface, that is at a depth of 5 cm at US-Wkg, 2 cm at US-Ton and 2 cm at US-WBW. The height above ground ($z$ in equation 4) was set to the flux tower height, which was 6.4 m at US-Wkg, 23 m at US-Ton and 36.9 m at US-WBW. At US-Wkg, the dry soil thermal inertia ($I_{ds}$ in equation 3) was computed according to Wang et al. (2010) as the regression coefficient between diurnal variations in the ground heat flux and surface temperature (830 J m$^{-2}$ K$^{-1}$ s$^{-1/2}$). Because these data were unavailable at US-Ton and US-WBW and given that the model showed little sensitivity to the soil thermal inertia, the dry soil thermal inertia was set equal to 800 J m$^{-2}$ K$^{-1}$ s$^{-1/2}$ for these two sites. In the MEP model of transpiration, the leaf surface temperature was assumed equal to the air temperature measured above the canopy. Since no measurements of the leaf surface specific humidity were available, we used the specific humidity of the air as a proxy and calculated it from air temperature and relative humidity measurements using the Clausius-Clapeyron equation. To combine the MEP models of evaporation and transpiration (equation 13), we computed the vegetation index (equation 14) using the AVHRR 7-day composite NDVI (https://lta.cr.usgs.gov/NDVI). NDVI time series are notoriously noisy because of varying atmospheric conditions and sensor viewing angles (Hird and

McDermid, 2009) and we therefore smoothed the time series by applying a 60-day moving average. Based on Montandon and Small (2008), the NDVI signal for bare soil ($NDVI_0$) was set to 0.2 and the NDVI signal for full vegetation cover varied according to the land cover, with $NDVI_\infty = 0.61$ for grassland, $NDVI_\infty = 0.69$ for woody savanna, and $NDVI_\infty = 0.85$ for deciduous broadleaf forest.

**Table 3. Implementation of the MEP model of total terrestrial evaporation in this study**

| Model | Variable | Definition | Implementation |
|---|---|---|---|
| MEP-E$_s$ | $R_n$ | net radiation | net radiation measurements |
| | $T_{ss}$ | surface temperature | soil temperature measurements nearest to the surface |
| | $q_{ss}$ | surface specific humidity | equation 10; computed from soil water content (supplied by HGS), porosity (Table 2) and surface temperature measurements |
| | $I_s$ | soil thermal inertia | set as a constant in this study |
| MEP-E$_t$ | $R_n$ | net radiation | net radiation measurements |
| | $T_{ls}$ | surface temperature | air temperature measurements at top of the tower |
| | $q_{ls}$ | surface specific humidity | computed from air temperature and relative humidity measurements at the top of the tower |
| | $\eta_s$ | water stress factor | equation 12; computed from the soil water content (supplied by HGS) and the soil water content at wilting point and field capacity (Table 2) |
| MEP-E | $f_{veg}$ | vegetation index | equation 14; computed from NDVI data |

### 3.4.2 HGS with Penman-Monteith (HGS-PM)

We compared HGS-MEP to simulations of terrestrial evaporation and soil water content performed with the standalone HGS model, in which the Kristensen and Jensen (1975) model is implemented, as described in section 3.2. We used the same soil
and root properties defined for the HGS-MEP simulations (Table 2) and the only difference lied in the terrestrial evaporation model. In HGS, transpiration and evaporation are set to occur simultaneously. However, this approach led to a large overestimation of terrestrial evaporation. In the present study, we modified the implementation of the model and evaporation took place only when $f_1(LAI) = 0$. As for the HGS-MEP model implementation, interception was not considered and $E_c$ was set to zero. The maximum and minimum evaporation limiting saturation (equation 18) were set to 0.1 ($\theta_{e1} = 0.1\theta_s$) and 0.5
($\theta_{e2} = 0.5\theta_s$) (Verbist et al., 2012). Potential evaporation ($E_p$), that is terrestrial evaporation from a saturated land surface, was computed with the Penman-Monteith equation, a physically based model where, similar to the MEP model, the predicted terrestrial evaporation is constrained by available energy at the surface:

$$E_p = \frac{1}{\lambda}\frac{\Delta(R_n - G) + \frac{\rho_a c_p(e_s - e_a)}{r_a}}{\Delta + \gamma\left(1 + \frac{r_s}{r_a}\right)} \tag{20}$$

where $\Delta$ is the slope of the saturation vapor pressure curve (kPa K$^{-1}$), $e_s$ is the saturation vapor pressure (kPa), $e_a$ is the actual
vapor pressure (kPa), $\gamma$ is the psychrometric constant (kPa K$^{-1}$), $r_s$ is the surface resistance (m s$^{-1}$) and $r_a$ is the aerodynamic resistance (m s$^{-1}$). When $f_1(LAI) > 0$, the surface resistance ($r_s$) was set according to the vegetation lookup table in the Noah land surface model (US-Wkg = 40 s m$^{-1}$, US-Ton = 70 s m$^{-1}$, US-WBW = 100 s m$^{-1}$ in Kumar et al., 2011) and the canopy aerodynamic resistance ($r_{ac}$) was computed as (Thom, 1975):

$$r_{ac} = \frac{1}{\kappa^2 u}\left[\ln\left(\frac{z - d_0}{z_{0m}}\right)\ln\left(\frac{z - d_0}{z_{0v}}\right)\right] \tag{21}$$

where $u$ is the wind speed (m s$^{-1}$), $z$ is the wind speed measurement height (m), $d_0$ is the zero-plane displacement height (m), $z_{0m}$ is the roughness height for momentum transfer (m) and $z_{0v}$ is the roughness height for water vapor transfer (m). Equation 21 was derived for neutral atmospheric conditions but has also been successfully used to model terrestrial evaporation over a wide range of conditions (Ershadi et al., 2014). Roughness heights were estimated as a fraction of the vegetation height, $h$ (m) (Brutsaert, 1982):

$$d_0 = 0.66h \tag{22}$$

$$z_{0m} = 0.1h \tag{23}$$

$$z_{0v} = 0.1z_{0m} \tag{24}$$

When $f_1(LAI) = 0$, the surface resistance ($r_s$) was set to 999 s m$^{-1}$, the value associated with a barren/sparsely vegetated land cover in the Noah lookup table (Kumar et al., 2011), and the substrate aerodynamic resistance ($r_{as}$) was computed as (Shuttleworth and Wallace, 1985):

$$r_{as} = \frac{1}{\kappa^2 u}\left[\ln\left(\frac{z}{z_0'}\right)\ln\left(\frac{d_0+z_{0m}}{z_0'}\right)\right] \tag{25}$$

where $z_0'$ is the roughness length of the soil (m) which was set to 0.01 m (Shuttleworth and Wallace, 1985).

The $f_1(LAI)$ function used to describe temporal changes in vegetation (equation 16) was computed by rescaling the vegetation index $f_{veg}$, an input to the HGS-MEP model, between 0 and 1.

### 3.5 Model setup

As a proof-of-concept, we set up the coupled HGS-MEP and HGS-PM models to perform one-dimensional soil column simulations to evaluate the capability of both models to simulate water fluxes (terrestrial evaporation) and storage (soil moisture). We represented the soil column with a fine (1 cm) vertical resolution and set the soil column depth to either 1 or 1.5 m in order to capture the entire root zone at each site. We assigned uniform soil properties throughout the soil column since there was no available data to describe the vertical distribution of soil material with depth. Simulations spanned five years at US-Wkg and US-Ton and 2.5 years at US-WBW given data were not available for a longer period. We used soil water content measurement available at different depths to set initial subsurface conditions and we used linear interpolation to assign initial conditions at depths without measurements. As for initial surface water conditions, we assumed an initial surface water depth of zero given the soil was not fully saturated. For boundary conditions, we supplied the model with gap-filled (REddyProc, Reichstein et al., 2005) measurements of precipitation, net radiation, air temperature, relative humidity and soil temperature at a 30-minute time step. At the surface, we applied a critical depth boundary condition that allows water to leave the model domain via overland flow. At the bottom of the soil column, we applied a free drainage boundary condition.

### 3.6 Model performance

We evaluated the performance of models using a series of metrics comparing observed and simulated values of terrestrial evaporation and soil water content at the three sites. First, we computed the root mean square error (RMSE) to assess the mean difference between observed and simulated values. Second, we computed the Nash-Sutcliffe efficiency (NSE), where a value of one indicates a perfect agreement between the model and observations and a negative value indicates that average value of observations offers a better predictor than the model. The RMSE and NSE are not independent metrics of performance given that the NSE is a standardized measure of the mean square error. Still, we chose to report the two commonly used metrics as they provide an assessment of performance in absolute (RMSE) and relative (NSE) terms. Third, we computed the normalized benchmark efficiency (BE), which is analog to the NSE, but rather compares the model output to a simple benchmark model, in this case, the interannual mean value for every calendar day (Schaefli and Gupta, 2007). Fourth, we computed the coefficient of determination ($R^2$) that describes the proportion of the total variance in observations

explained by the model. Finally, we computed the percent bias (PBIAS) to assess the average tendency of simulated values to be larger (positive bias) or smaller (negative bias) than observations. Equations of performance metrics are listed in Table S1.

We calculated performance metrics at half-hourly and daily time scales for terrestrial evaporation and at a daily time scale for soil water content. When assessing the ability of the models to simulate terrestrial evaporation, we first calculated
performance metrics on the entire time series and second, assessed how well the models performed under water-limited conditions. To do so, we computed the monthly aridity index, that is the ratio between precipitation and PET, where PET was calculated with the Penman-Monteith equation (eq. 20). Using monthly values of the aridity index, we then calculated performance metrics for dry periods (P/PET < 0.4 at US-Wkg and P/PET < 1 at US-Ton and US-WBW) and wet periods (P/PET ≥ 0.4 at US-Wkg and P/PET ≥ 1 at US-Ton and US-WBW). As US-Wkg is located in a semi-arid climate, we used a
different threshold (0.4) to define water-limited conditions as the monthly aridity index remained below one for the entire study period.

## 4 Results

### 4.1 Model performance under a semiarid climate (US-Wkg)

During the study period (2010–2014), the mean annual precipitation at US-Wkg varied between 264 and 415 mm, with 2013
and 2014 being the driest and wettest years, respectively (Figure 3a). Between January and June, pre-monsoon precipitation was relatively low (≤ 35 mm), with the exception of 2010, which saw 110 mm of rain in the first six months of the year. Precipitation was concentrated during the monsoon periods and, with the exception of 2010, nearly 80% occurred between July and September. Figure 3b shows that the HGS-MEP model provided a realistic simulation of soil moisture at 15 cm depth (RMSE = 0.04 $m^3$ $m^{-3}$; NSE = 0.30; Table 4) and was able to capture the sharp rise in soil moisture at the start of the
monsoon period in July – note that soil moisture observations were missing for the 2010/217 to 2011/365 period. Overall, soil moisture at 15 cm depth was generally overestimated (PBIAS = 19%; Table 4) with HGS-MEP, particularly the lower values outside the monsoon period. For example, the observed annual minimum in soil water content was 0.06 $m^3$ $m^{-3}$ between 2012 and 2014, while modeled soil moisture did not fall below 0.12 $m^3$ $m^{-3}$. A similar overestimation was also observed for the upper ($z$ = 5 cm) and lower ($z$ = 30 cm) soil layers (Figure S1).

Figure 3c shows that the HGS-MEP model also performed well in simulating the daily mean terrestrial evaporation (RMSE = 0.31 mm $day^{-1}$; NSE = 0.88; Table 4). The HGS-MEP model reproduced the seasonal pattern in terrestrial evaporation and captured the sharp increase associated with the increased soil water availability from July to September. Peak terrestrial evaporation was slightly underestimated by HGS-MEP (PBIAS = –10%; Table 4). For example, the observed annual maximum ranged between 4.8 and 5.8 mm $day^{-1}$, while the modeled annual maximum did not exceed 4.0 mm $day^{-1}$. The
HGS-MEP model was also able to simulate the increase in terrestrial evaporation following precipitation events outside the monsoon months. However, the pre-monsoon months were particularly wet in 2010 and terrestrial evaporation simulated by HGS-MEP was underestimated with a modeled average of 0.5 mm $day^{-1}$ between January and May compared to an observed average of 0.9 mm $day^{-1}$. At US-Wkg, terrestrial evaporation was dominated by evaporation, particularly during the dry season during which it represented 69% of the terrestrial evaporation simulated by HGS-MEP (Figure 4a). During the
monsoon period when vegetation activity is concentrated (July to October), evaporation decreased in importance and, on average, accounted for 60% of the total terrestrial evaporation simulated by HGS-MEP, while transpiration represented 40% of total terrestrial evaporation. These proportions varied from year to year. For example, 2014 was the wettest year during the study period and modeled transpiration represented 46% of total terrestrial evaporation during the monsoon period. In contrast, modeled transpiration represented 28% of total terrestrial evaporation during the monsoon period in 2012, although
terrestrial evaporation was overall underestimated by the HGS-MEP model that year. Overall, the HGS-MEP model outperformed the HGS-PM model for the simulation of daily terrestrial evaporation and soil moisture (Figure 3b). Indeed, performance metrics at the daily time scale show a large decline in the performance of HGS-PM during wet periods (NSE = 0.04) compared to dry periods (NSE = 0.64; Table 5). At the opposite, the performance of HGS-MEP was comparable for dry

(NSE = 0.86) and wet periods (NSE = 0.85). Moreover, in-between monsoon periods, terrestrial evaporation was underestimated by HGS-PM (PBIAS = –14%; Table 4) and the model was unable to catch terrestrial evaporation pulses following rain events (e.g. 2013, Figure 3c). As a result, soil moisture was generally overestimated (PBIAS = 27%) throughout the simulation period. At the diurnal scale, the HGS-MEP model reproduced well subdaily variations in terrestrial evaporation at US-Wkg, with an increase in the morning, peak value around 12:00, a decrease in the afternoon, and values close to zero during the night (Figure 5a). In the HGS-MEP simulation, the daily maximum was generally overestimated with an average of 0.12 mm h$^{-1}$ at 12:00, compared to an observed average of 0.10 mm h$^{-1}$. On the other hand, morning and afternoon values were slightly underestimated, by about 0.01 mm h$^{-1}$ (Figure 5a). While the HGS-PM did not overestimate the daily maximum (Figure 5a), the HGS-MEP model overall outperformed HGS-PM for the simulation of half-hourly terrestrial evaporation, as shown by the four performance metrics (RMSE, NSE, R², PBIAS, Table 6).

**Table 4. Performance of the HGS-MEP and HGS with Penman-Monteith (HGS-PM) models when simulating a) daily mean soil water content (SWC) at a 15- or 20-cm depth and b) daily mean total terrestrial evaporation (E) as represented by the root mean square error (RMSE), Nash-Sutcliffe Efficiency (NSE), benchmark efficiency (BE), coefficient of determination (R²) and percentage bias (PBIAS).**

| | RMSE | | NSE | | BE | | R² | | PBIAS | |
|---|---|---|---|---|---|---|---|---|---|---|
| | HGS-MEP | HGS-PM | HGS-MEP | HGS-PM | HGS-MEP | HGS-PM | HGS-MEP | HGS-PM | HGS-MEP | HGS-PM |
| **a) SWC** | (m$^3$ m$^{-3}$) | | | | | | | | (%) | |
| US-Wkg | 0.04 | 0.05 | 0.30 | –0.10 | –0.35 | –1.13 | 0.62 | 0.54 | 19 | 27 |
| US-Ton | 0.03 | 0.04 | 0.92 | 0.88 | 0.60 | 0.46 | 0.94 | 0.89 | –5 | 0 |
| US-WBW | 0.05 | 0.05 | 0.61 | 0.51 | –0.22 | –0.54 | 0.74 | 0.66 | –6 | –1 |
| **b) E** | (mm day$^{-1}$) | | | | | | | | (%) | |
| US-Wkg | 0.31 | 0.58 | 0.88 | 0.57 | 0.48 | –0.86 | 0.89 | 0.65 | –10 | –14 |
| US-Ton | 0.43 | 0.55 | 0.73 | 0.56 | 0.11 | –0.45 | 0.77 | 0.70 | –14 | –25 |
| US-WBW | 0.71 | 0.74 | 0.65 | 0.62 | –1.70 | –1.93 | 0.68 | 0.69 | 11 | –23 |

**Table 5. Performance of the HGS-MEP and HGS with Penman-Monteith (HGS-PM) models when simulating daily mean total terrestrial evaporation (E) during dry and wet periods as represented by the root mean square error (RMSE), Nash-Sutcliffe Efficiency (NSE), coefficient of determination (R²) and percentage bias (PBIAS). n represents the number of days.**

| | n | RMSE | | NSE | | R² | | PBIAS | |
|---|---|---|---|---|---|---|---|---|---|
| | | HGS-MEP | HGS-PM | HGS-MEP | HGS-PM | HGS-MEP | HGS-PM | HGS-MEP | HGS-PM |
| | | (mm day$^{-1}$) | | | | | | (%) | |
| **US-Wkg** | | | | | | | | | |
| dry (P/PET<0.4) | 1518 | 0.29 | 0.46 | 0.86 | 0.64 | 0.87 | 0.68 | –11 | –15 |
| wet (P/PET≥0.4) | 308 | 0.39 | 0.98 | 0.85 | 0.04 | 0.83 | 0.47 | –9 | –12 |
| **US-Ton** | | | | | | | | | |
| dry (P/PET<1) | 1340 | 0.43 | 0.56 | 0.76 | 0.59 | 0.81 | 0.72 | –14 | –24 |
| wet (P/PET≥1) | 486 | 0.44 | 0.49 | 0.37 | 0.20 | 0.58 | 0.61 | –10 | –27 |
| **US-WBW** | | | | | | | | | |
| dry (P/PET<1) | 489 | 0.78 | 0.87 | 0.54 | 0.42 | 0.57 | 0.57 | 6 | –23 |
| wet (P/PET≥1) | 425 | 0.61 | 0.54 | 0.57 | 0.66 | 0.65 | 0.71 | 26 | –24 |

**Table 6. Performance of the HGS-MEP and HGS with Penman-Monteith (HGS-PM) models when simulating half-hourly mean total terrestrial evaporation as represented by the root mean square error (RMSE), Nash-Sutcliffe Efficiency (NSE), coefficient of determination (R²) and percentage bias (PBIAS).**

|  | RMSE | | NSE | | R² | | PBIAS | |
|---|---|---|---|---|---|---|---|---|
|  | HGS-MEP | HGS-PM | HGS-MEP | HGS-PM | HGS-MEP | HGS-PM | HGS-MEP | HGS-PM |
|  | (mm h$^{-1}$) | | | | | | (%) | |
| US-Wkg | 0.05 | 0.06 | 0.65 | 0.48 | 0.72 | 0.58 | 3 | –9 |
| US-Ton | 0.06 | 0.06 | 0.53 | 0.55 | 0.61 | 0.60 | –3 | –18 |
| US-WBW | 0.10 | 0.08 | 0.47 | 0.62 | 0.57 | 0.64 | 23 | –18 |

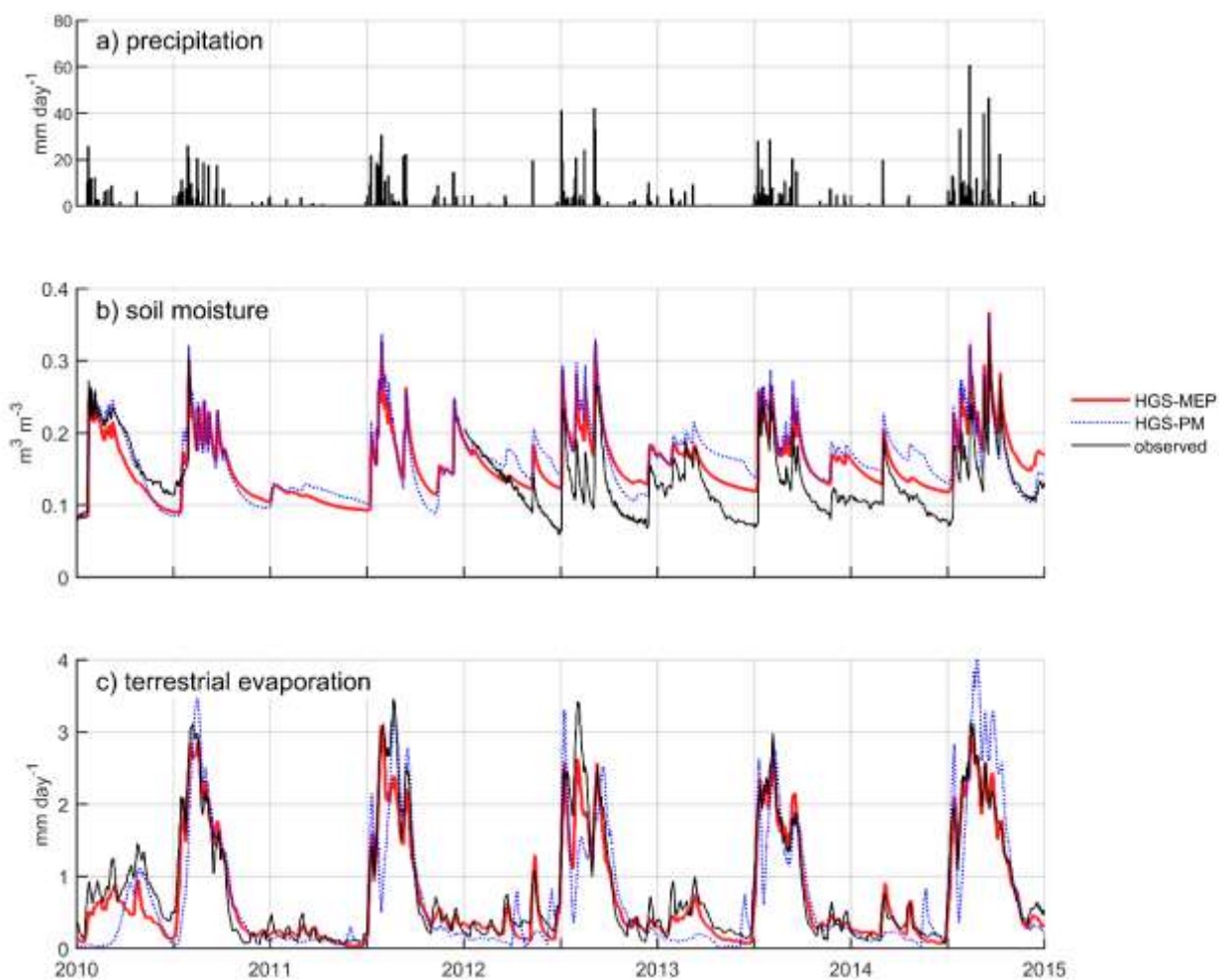

**Figure 3. a) Precipitation, as well as observed and modeled b) daily mean soil water content at a depth of 15 cm and c) 10-day moving average of terrestrial evaporation at US-Wkg (climate: semiarid, vegetation: grassland).**

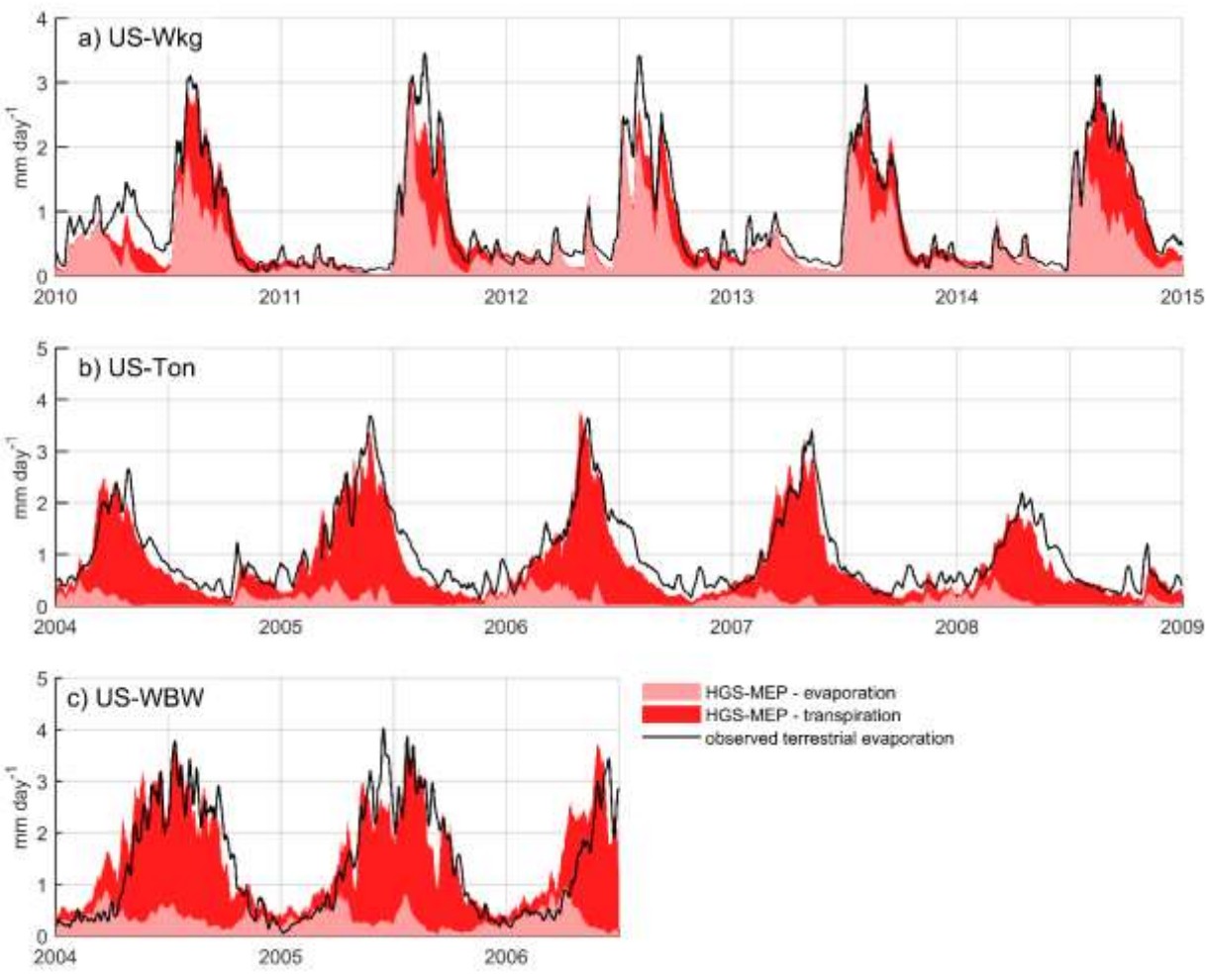

**Figure 4. Ten-day moving average of observed terrestrial evaporation and modeled terrestrial evaporation partitioned as transpiration and evaporation at a) US-Wkg (climate: semiarid, vegetation: grassland), b) US-Ton (climate: Mediterranean, vegetation: woody savanna) and c) US-WBW (climate: temperate, vegetation: deciduous broadleaf forest).**

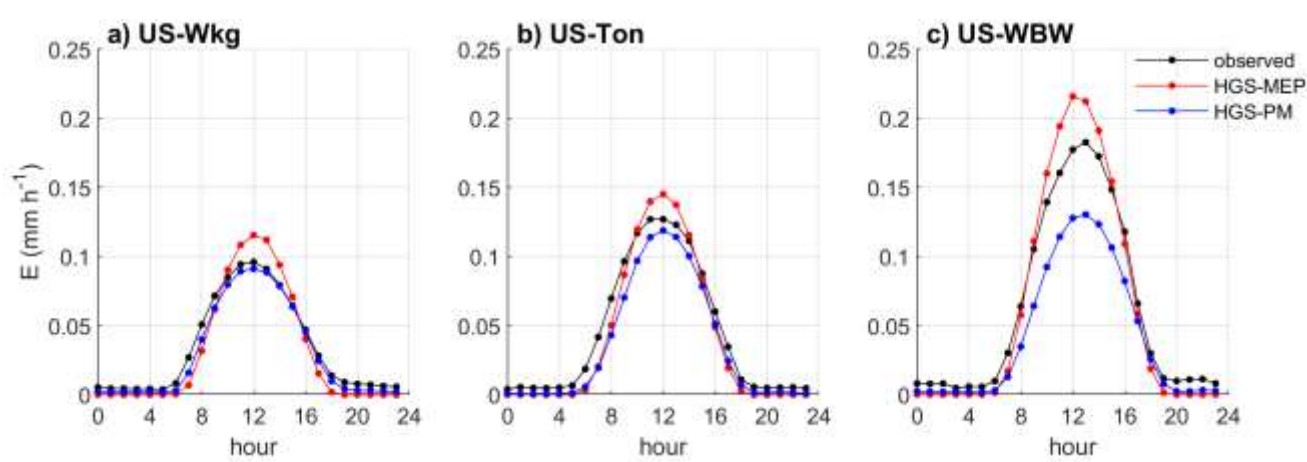

**Figure 5. Observed and modeled hourly average terrestrial evaporation at US-Wkg (climate: semiarid, vegetation: grassland), b) US-Ton (climate: Mediterranean, vegetation: woody savanna) and c) US-WBW (climate: temperate, vegetation: deciduous broadleaf forest).**

## 4.2 Model performance under a Mediterranean climate (US-Ton)

During the study period (2004–2008), mean annual precipitation at US-Ton ranged between 371 and 782 mm, with most precipitation concentrated during winter months from October to May (Figure 6a). The first two winters (2004–2005 and 2005–2006) of the study period were particularly wet (precipitation = 717 mm and 882 mm), while the following years experienced near-normal precipitation (385 mm and 392 mm). As shown in Figure 6b, the HGS-MEP model simulated soil moisture exceptionally well at a 20-cm depth (RMSE = 0.03 $m^3$ $m^{-3}$; NSE = 0.92; Table 4) and was able to reproduce the decrease in soil moisture as precipitation stops in summer (Figure 6b). During the wet winter months, the HGS-MEP model slightly underestimated soil moisture (PBIAS = –5%) but still captured the increase in soil moisture during this period. Near the surface (z = 0 cm, Figure S2a), soil moisture was generally underestimated by HGS-MEP during the dry summer months: observed soil water content slowly decreased until it generally reached a minimum of 0.04 – 0.05 $m^3$ $m^{-3}$ at the end of the season, while modeled soil moisture dropped rapidly to reach a minimum value (0.03 $m^3$ $m^{-3}$) close to the residual water content (0.02 $m^3$ $m^{-3}$; Table 2). For the deeper soil layers ($z$ = 50 cm), the HGS-MEP model generally overestimated soil moisture during dry summer months with a modeled minimum of 0.17 $m^3$ $m^{-3}$ compared to an observed minimum of 0.14 $m^3$ $m^{-3}$ (Figure S2c).

When simulating terrestrial evaporation , the HGS-MEP model performed well (RMSE = 0.43 mm $day^{-1}$; NSE = 0.73) although terrestrial evaporation was underestimated (PBIAS = –14%; Table 4). The HGS-MEP model performed particularly well for winter months and was able to capture the increase in terrestrial evaporation from about 0.4 mm $day^{-1}$ (modeled average terrestrial evaporation in September) to 3.3 mm $day^{-1}$ (modeled average annual maximum terrestrial evaporation) as water became more available. However, following those winter months, as precipitation stopped, terrestrial evaporation simulated by HGS-MEP was underestimated during the first half of the dry period (June – July). For example, the underestimation in terrestrial evaporation during the summer of 2005 and 2006 (i.e. from the last day of precipitation in May or June to the end of September) amounted respectively to a cumulative difference of 8.1 and 6.8 mm between observed and modeled terrestrial evaporation. During the dry summer months, low soil moisture availability near the surface (Figure S2a) limited evaporation simulated by HGS-MEP and modeled transpiration accounted on average for 94% of total terrestrial evaporation (Figure 4b). Evaporation increased during the monsoon period and, on average, terrestrial evaporation simulated by HGS-MEP was made up of 17% of evaporation and 83% of transpiration. Overall, the HGS-PM model performed well at US-Ton at a daily time scale, although the HGS-MEP performed slightly better. As opposed to HGS-MEP, the HGS-PM had difficulties capturing the onset of the wet winter period and the increase in soil moisture occurred ahead of time (Figure 6b). Indeed, we observed a decline in the performance of HGS-PM during wet periods (NSE = 0.20) compared to dry periods (NSE = 0.59; Table 5). We observed a similar pattern with HGS-MEP, but not as marked with a NSE of 0.37 during wet periods compared to 0.76 during dry periods (Table 5). With HGS-MEP and HGS-PM, terrestrial evaporation was underestimated, both during wet and dry periods (–27% ≤ PBIAS ≤ –10%, Table 5). This underestimation was particularly important in June and July when precipitation stopped (Figure 6c). At the diurnal scale, both models performed similarly well (NSE = 0.53 for HGS-MEP and NSE = 0.55 for HGS-PM), although HGS-PM led to a more important bias (PBIAS = –18%) than HGS-MEP (PBIAS = –3%; Table 6). Throughout the day, both models underestimated terrestrial evaporation in the morning and afternoon, while the daily maximum was generally overestimated by HGS-MEP and slightly underestimated by HGS-PM (Figure 5b).

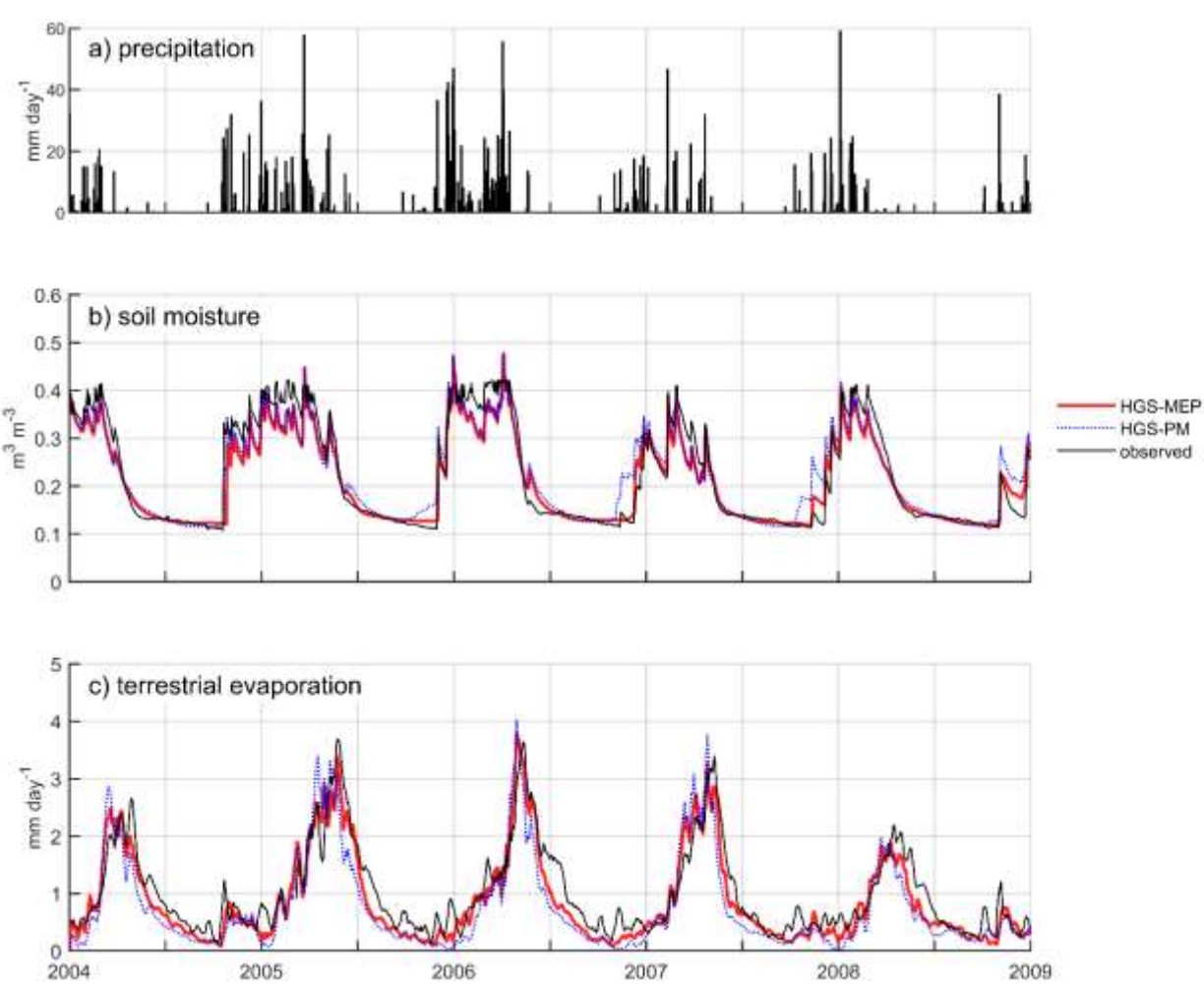

**Figure 6. a) Precipitation, as well as observed and modeled b) daily mean soil water content at a depth of 20 cm and c) 10-day moving average of terrestrial evaporation at US-Ton (climate: Mediterranean, vegetation: woody savanna).**

### 4.3 Model performance under a temperate climate (US-WBW)

In the simulation period, 2004 was a relatively wet year (annual precipitation = 1600 mm), while 2005 was relatively dry (annual precipitation = 995 mm; Figure 7a). In Figure 7b, the HGS-MEP model provided a realistic simulation of soil moisture (RMSE = 0.05 $m^3$ $m^{-3}$; NSE = 0.61; Table 4) and captured the decrease in soil moisture associated with increased vegetation activity during the summer. However, soil moisture at a 20 cm depth was generally overestimated by HGS-MEP during the summer and modeled soil water content did not fall below 0.1 $m^3$ $m^{-3}$ while the observed soil water content reached an annual minimum of 0.08 $m^3$ $m^{-3}$ and 0.05 $m^3$ $m^{-3}$ in 2004 and 2005. At the opposite, soil moisture during the winter was generally underestimated by HGS-MEP with a modeled average soil water content of 0.21 $m^3$ $m^{-3}$ between January and April in comparison with an observed average of 0.26 $m^3$ $m^{-3}$. Overall, a similar pattern (overestimation of soil moisture in the summer and underestimation in the winter) was observed in soil moisture simulations near the surface ($z = 5$ cm; Figure S3a). As for deeper soil layers ($z = 60$ cm), soil moisture was systematically underestimated, although this could be due to changes in soil properties given there is an upper shift in observed soil moisture values compared to upper soil layers (Figure S3c).

In Figure 7c, the HGS-MEP model performed well and reproduced the seasonal pattern in terrestrial evaporation (RMSE = 0.71 mm day$^{-1}$; NSE = 0.65; Table 4). Capturing the onset of vegetation activity was, however, challenging for the model and, early in the summer, terrestrial evaporation was generally overestimated (PBIAS = 11%). On two occasions, terrestrial

evaporation was considerably underestimated by the HGS-MEP model: in June 2005, observed terrestrial evaporation reached a peak of 3.9 mm day$^{-1}$, while modeled terrestrial evaporation was about 2.5 mm day$^{-1}$ and in September 2005, observed terrestrial evaporation reached 2.8 mm day$^{-1}$, while modeled terrestrial evaporation went below 1 mm day$^{-1}$ (values refer to the 10-day moving average presented in Figure 4c). This underestimation in terrestrial evaporation is linked to the water stress factor. In both cases, the water stress factor fell below 0.5 for a few days, which largely reduced modeled transpiration. Evaporation simulated by HGS-MEP was relatively constant throughout the year with an average of 0.4 mm day$^{-1}$, although in the fall 2005, following a particularly dry summer, modeled evaporation dropped to less than 0.1 mm day$^{-1}$. Transpiration simulated by HGS-MEP accounted on average for 57% of total terrestrial evaporation between October and April, but this proportion increased considerably during the summer. Indeed, between May and September, modeled transpiration represented on average 87% of total terrestrial evaporation. Overall, the HGS-MEP model outperformed HGS-PM, although, as shown by negative BE values, both models had less explanatory power than a simple benchmark model that captures seasonality. The two models performed similarly during dry and wet periods, with only a slight decline in performance at HGS-PM during dry periods in the summer (NSE = 0.42) compared to wet periods in the winter (NSE = 0.66; Table 5). During summer months, the HGS-PM largely underestimated terrestrial evaporation (PBIAS = –23%), particularly during the dry 2005 year (Figure 7c). As a result, soil moisture was overestimated by HGS-PM, and much more so than with HGS-MEP (Figure 7b). During winter months, soil moisture was generally underestimated, although a similar pattern was observed for HGS-MEP. At the diurnal scale, the HGS-PM (NSE = 0.62) performed better than HGS-MEP (NSE = 0.47) when simulating half-hourly terrestrial evaporation, although both models had an important bias, positive for HGS-MEP (PBIAS = 23%) and negative for HGS-PM (PBIAS = –18%; Table 6). This bias is particularly reflected in the simulation of the daily maximum which was overstimated by HGS-MEP  and largely underestimated by HGS-PM (Figure 5c). Indeed, the daily maximum observed mid-day at US-WBW is 0.18 mm h$^{-1}$, while it reached 0.22 mm h$^{-1}$ with HGS-MEP and 0.13 mm h$^{-1}$ with HGS-PM.

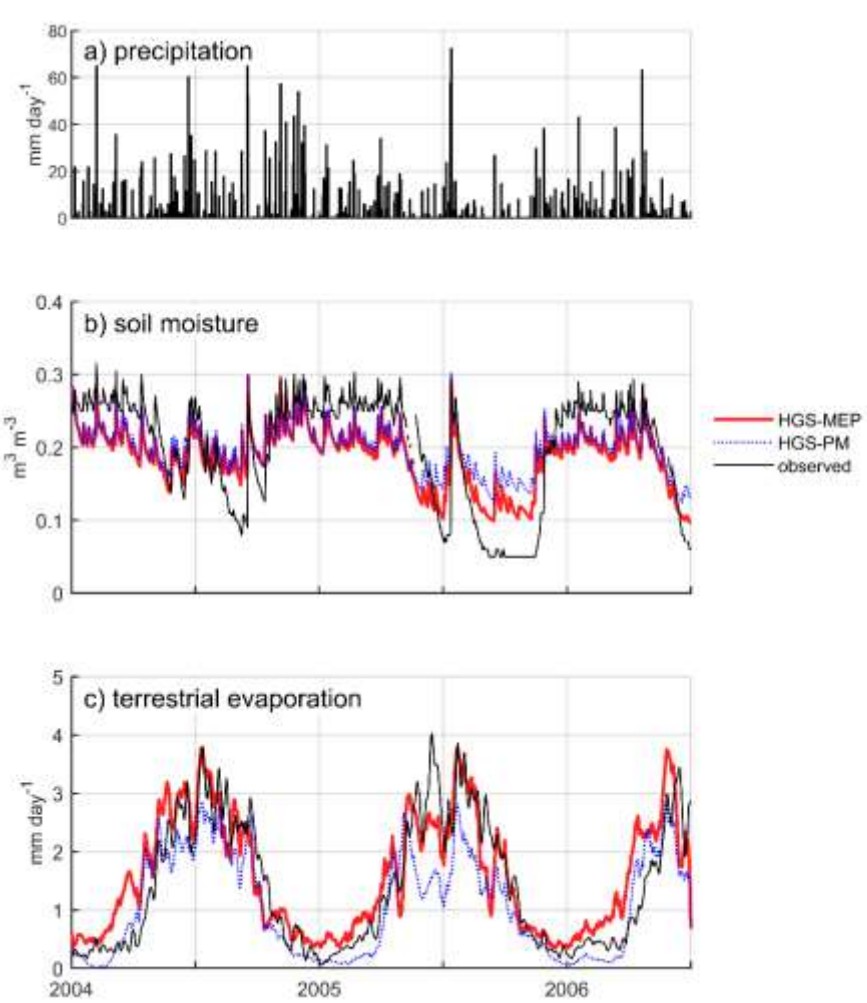

**Figure 7. a) Precipitation, as well as observed and modeled b) daily mean soil water content at a depth of 20 cm and c) 10-day moving average of terrestrial evaporation at US-WBW (climate: temperate, vegetation: deciduous broadleaf forest).**

## 5 Discussion

### 5.1 Performance of the HGS-MEP model

#### 5.1.1 HGS-MEP vs HGS-PM

At the daily time scale, the HGS-MEP model outperformed HGS-PM when simulating terrestrial evaporation at the three study sites, which translated into improved performance for soil moisture modeling as well (Table 4). Notably, we observed a weak performance of HGS-PM during wet periods at US-Wkg (P/PET≥0.4) and US-Ton (P/PET≥1; Table 5). At US-Wkg, this meant that HGS-PM failed to capture the annual maximum terrestrial evaporation which has important implications for the water budget. At the opposite, the weaker performance at US-Ton by HGS-PM, and to a lesser degree by HGS-MEP, meant that the two models struggle to describe the minimum rates of terrestrial evaporation. At US-WBW, both models had a comparable performance with consistent results for HGS-MEP for wet (P/PET≥1) and dry periods, while HGS-PM performed better during wet periods than dry periods (Table 5).

At the semiarid site US-Wkg (Figure 3), a better handling of terrestrial evaporation partitioning can explain the superior performance of HGS-MEP compared to HGS-PM, at both daily and half-hourly time scales. Indeed, the HGS-MEP model

assesses transpiration and evaporation independently one from another. In the Kristensen and Jensen (1975) model in HGS, transpiration and evaporation are instead derived from a single PET value, although we could realistically expect potential transpiration and evaporation to differ from one another. In fact, Shuttleworth and Wallace (1985) have proposed a two-layer configuration of the Penman-Monteith model which allows the definition of different resistance values for the soil and canopy layers. However, HGS remains limited to the definition of a single PET value at the moment. We encountered a large overestimation of terrestrial evaporation when deriving transpiration and evaporation from a single PET value and we set up the HGS model to only allow bare soil evaporation in the absence of active vegetation ($f_1(LAI) = 0$). As a result, the HGS-PM model performed weakly at US-Wkg, where soil evaporation makes up more than 50% of terrestrial evaporation according to observations (Moran et al., 2009). Soil moisture modeling by HGS-MEP also proved somewhat challenging at US-Wkg (NSE = 0.30, BE = –0.35; Table 4). However, in the absence of parameter calibration, we consider this performance more than acceptable given the challenges of modeling water fluxes under arid climates. Soil moisture was generally overestimated by HGS-MEP during dry periods (Figure 3) and further work on the definition of the wilting point in arid climates could help improve the performance of the model. Indeed, wilting can occur at a lower threshold than –1.5 MPa (–4 to –2 MPa, Baldocchi et al., 2004) for vegetation having evolved under an arid climate.

At the Mediterranean site US-Ton, daily terrestrial evaporation was underestimated by HGS-MEP (PBIAS = –14%, Table 4), particularly during the second half of the year, as reduced water supply led to a decline in terrestrial evaporation (Figure 5). While the HGS-MEP model simulates soil moisture very well at a depth of 20 cm (Figure 6), it tended to underestimate soil moisture close to the surface (Figure S2a), where the largest proportion of roots is found according to the vertical root distribution defined by HGS (cubic decay distribution between the surface and the maximum root depth). Given that the water stress factor ($\eta_s$) is computed from the weighted average soil water content over the root zone (equation 12), this underestimation of soil moisture near the surface translated into an overestimation of the reduction in transpiration resulting from water stress. We also investigated if the issue of water stress overestimation could be due to a misdefinition of the maximum rooting depth, as trees under a Mediterranean climate have been found to access water from deep soil layers or groundwater (Miller et al., 2010). However, we also simulated terrestrial evaporation with the stand-alone MEP model using soil moisture observations, thus avoiding the overestimation of water stress near the surface, and instead found a large overestimation of terrestrial evaporation (Figure S4). These results suggest that increasing the rooting depth to increase access to water resources would likely not improve the simulation of terrestrial evaporation. Instead, uncertainty relative to the definition of the vertical root distribution (as opposed to the maximum rooting depth) or, as previously discussed, with the definition of water stress points (wilting point and field capacity) may explain the challenge of simulating terrestrial evaporation under water-limited conditions at US-Ton.

At the half-hourly time scale, both HGS-MEP and HGS-PM showed lower performance than at a daily time scale when simulating terrestrial evaporation. For example, the NSE varied between 0.56 and 0.88 at a daily time scale (Table 4), while it varied between 0.47 and 0.65 at a half-hourly time scale (Table 6). At a half-hourly time scale, no model is distinctly superior from one another: HGS-MEP performed better at US-Wkg, HGS-PM performed better at US-WBW and both models performed similarly at US-Ton (Table 6). Still, we observed a distinct pattern where peak terrestrial evaporation during the day was generally overestimated by HGS-MEP and underestimated by HGS-PM (Figure 5). A known issue of energy imbalance, particularly at subdaily time scales, is associated with eddy covariance measurements (Leuning et al., 2012). This issue typically leads to an underestimation in observations of terrestrial evaporation and could in part explain the apparent issue of overestimation of peak values by HGS-MEP. On the other hand, the negative bias of HGS-PM at the half-hourly time scale (–18% ≤ PBIAS ≤ –9%; Table 6) may actually be even more important when taking into account the energy imbalance issue.

Overall, the predictive ability of the MEP model is particularly noteworthy given that we did not rely on calibration and instead used a priori estimation of parameters describing the soil and vegetation. The MEP model thus offers a promising alternative to model hydrologic fluxes without relying on calibration (Wagener, 2007). We chose the Penman-Monteith as a benchmark against which to compare the MEP model as it is a physically based model that allows for a detailed

parameterization of vegetation. However, studies have shown that the Penman-Monteith leads to an underestimation of terrestrial evaporation under a contemporary climate (Ershadi et al., 2014) and an overestimation under climate change (Milly and Dunne, 2016). Other models could have been considered, although Hajji et al. (2018) demonstrated the superior performance of the MEP model compared to other models such as the modified Priestley-Taylor-Jet Propulsion Laboratory (PT-JPL) and the air-relative-humidity-based two-source model (ARTS).

### 5.1.2 HGS-MEP vs HGS using observed terrestrial evaporation as a forcing

Maheu et al. (2018) assessed HGS' skills to model soil moisture at the same AmeriFlux sites used in the present study. In Maheu et al. (2018) and the present study, the same input values were used to define soil and vegetation properties. Both studies also considered the same periods, with the exception of the Mediterranean site (US-Ton) where simulations were performed for different periods (2004-2008 in the present study vs 2008-2012 in Maheu et al., 2018) due to data availability. When modelling soil moisture with HGS, Maheu et al. (2018) used observed terrestrial evaporation as a forcing, thus reducing the uncertainty of simulating this flux and focusing the assessment of the model on subsurface processes and properties that control soil water content. These simulations thus offer a benchmark against which to compare the results of the present study and assess how much uncertainty is introduced by the simulation of terrestrial evaporation by the MEP model. At the three sites, the HGS simulations of soil moisture that used observed terrestrial evaporation as a forcing performed slightly better than simulations with HGS-MEP. Still, the performance was overall comparable between HGS with observed terrestrial evaporation (RMSE = 0.04 $m^3$ $m^{-3}$ and NSE between 0.39 and 0.86) and HGS-MEP (RMSE between 0.03 and 0.05 $m^3$ $m^{-3}$ and NSE between 0.30 and 0.92). At the semiarid site (US-Wkg), the HGS-MEP simulation of soil moisture had a greater overestimation bias (19%, Table 4) than the HGS simulation (10%, Table 4 in Maheu et al., 2018), which could be due to the underestimation of peak (2011 and 2012) or pre-monsoon (2010, 2012 and 2013) terrestrial evaporation (Figure 3). At the temperate site (US-WBW), simulations with HGS-MEP and HGS in Maheu et al. (2018) both showed the same pattern of underestimation of soil moisture in the winter and overestimation in the summer. This suggests that these biases are in good part associated with the definition of soil (hydraulic parameters) or vegetation (root distribution) properties rather than with the MEP model itself.

### 5.1.3 Partitioning of total terrestrial evaporation by HGS-MEP

In the present study, evaporation simulated by HGS-MEP represented on average 60% of total terrestrial evaporation at the semiarid site (US-Wkg) during the monsoon period (Figure 4a). These results are in line with experimental results from Moran et al. (2009) who estimated that, following the Lehmann lovegrass invasion, evaporation accounted for 55% of the total terrestrial evaporation during the growing season of a year with average precipitation. Results from the present study are also concordant with those of Scott and Biederman (2017) who found that, on average, evaporation represented 54% of growing-season total terrestrial evaporation at US-Wkg between 2004 and 2015. Using overstory and understory flux tower measurements at the Mediterranean site US-Ton, Miller et al. (2010) found that transpiration from trees dominate during the dry summer months and that understory evaporation (i.e. bare soil evaporation and transpiration from grasses and forbs) is close to zero. Our results are consistent with these observations and according to the HGS-MEP simulation, transpiration accounted on average for 94% of total terrestrial evaporation (Figure 4b). At the temperate site (US-WBW), soil evaporation, as measured by an understory flux tower, accounted for 16% of total terrestrial evaporation on an annual basis and for generally less than 8% of total terrestrial evaporation during the growing season (Wilson et al., 2001). In the present study, soil evaporation simulated by HGS-MEP amounted to 13% of total terrestrial evaporation during the growing season between May and September (Figure 4c), which agrees with these experimental estimates. Overall, the HGS-MEP model showed a good capability for the partitioning of total terrestrial evaporation under various climates (semiarid, Mediterranean and temperate); a conclusion that was also reached for a humid, energy-limited environment (Wang et al., 2017).

**5.2 Using the MEP model to integrate the energy budget in hydrological modeling: strengths and limitations**

The MEP model of land surface fluxes offers an effective means to implement coupled water and energy budget modeling for hydrological applications. First, the MEP model of terrestrial evaporation requires six input variables: net radiation, soil surface temperature, leaf surface temperature and specific humidity, vegetation index as well as soil water content, the latter
which can be supplied by a hydrological model. Thus, the MEP model eliminates the need for wind speed and surface roughness (input to the Penman model) or for vertical gradients of temperature and humidity (inputs to the aerodynamic method often implemented in land surface models). Second, the MEP model ensures, by design, the closure of the energy balance. As such, terrestrial evaporation simulated by the MEP model is constrained by available energy, which avoids issues of overestimation associated with the use of temperature-based PET models for hydrological projection. Third, the MEP
model relies on a small set of equations making it straightforward to implement with minimal computational needs. Land surface models also offer a means of implementing coupled water and energy budget modeling but, contrary to the MEP model, these models are generally complex and computationally heavy. Last, the explicit partitioning of total terrestrial evaporation into evaporation and transpiration is also a strength of the MEP model given that water feeding these two fluxes are drawn from different pools. Partitioning has particularly important implications for terrestrial evaporation modeling under
water-limiting conditions (e.g. arid environments, Kurc and Small, 2004) or under changing land cover conditions (Huxman et al., 2005) and the MEP model thus offers a tool to better represent these conditions in hydrological modeling.

While these four highlighted features make the MEP model a promising approach to couple water and energy budget modeling, certain limitations also need to be considered. First, the MEP model has mainly been tested with input data at a half-hourly time step. Although meteorological and climate data are increasingly available at a subdaily time step with the
continuing increase in the temporal resolution of reanalysis and climate projection datasets, additional tests would be needed to assess the applicability of the MEP model at a daily time scale. Second, soil water content is the key coupling variable between the MEP and hydrological models, which limits the choice of hydrological models to which the MEP model can be coupled to. For the moment, the choice of a hydrological model appears limited to physical models, with the drawback that these models are often computationally intensive as well as challenging in terms of parameterization. Indeed, the simulation
time for the one-dimensional soil columns in this study was greater than an hour with the HGS-MEP model. HGS is a relatively complex model and the MEP model has also been coupled to a soil moisture force-restore model with satisfying results (Huang and Wang, 2016, bias $< 0.01$ m$^3$ m$^{-3}$ and R$^2$ = 0.81). As for conceptual hydrological models, further investigation is needed to assess the possibility of a coupling with the MEP model. In the absence of information on soil water content, a method would be needed to derive the specific humidity at the soil surface ($q_{ss}$) as well as the water stress
factor ($\eta_s$) from the subsurface storage component of the conceptual model. Third, in its current form, the HGS-MEP model is still driven by dependent variables. For example, both net radiation and surface temperature are inputs to the model although incoming longwave radiation, a component of net radiation, is largely dependent on air temperature (used as a proxy for surface temperature $T_{ls}$). Moreover, soil surface temperature ($T_{ss}$) is an input to the model even though it is a function of the heat fluxes predicted by the model. In the present project, we have focused on the coupling of water fluxes between HGS
and MEP although, in the future, the two models could also be more closely coupled since thermal transport modeling is implemented within HGS (Brookfield et al., 2009). In addition to soil moisture information, the HGS model could supply information on soil surface temperature to the MEP model and thus eliminate its need as an input variable. For example, Huang and Wang (2016) simulated surface soil temperature and moisture using a force-restore model that relies on the MEP model to simulate the heat budget.

**6 Conclusion**

Using the MEP model of land surface fluxes, we proposed a simple approach to integrate energy budget modeling in hydrological models in order to improve the simulation of terrestrial evaporation. The MEP model requires six input variables (net radiation, soil surface temperature and specific humidity, leaf surface temperature and specific humidity and vegetation index) and ensures energy budget closure, which imparts a strong physical basis and avoids issues of

oversensitivity to air temperature associated with certain PET models. We coupled the MEP model to HGS, an integrated surface and subsurface hydrologic model. Without calibration, the coupled HGS-MEP model performed well in simulating soil water content and terrestrial evaporation at three AmeriFlux sites with varying climates (semiarid, Mediterranean, temperate). For both the simulation of daily soil moisture and terrestrial evaporation, HGS-MEP outperformed the standalone
HGS model where, as defined by the Kristensen and Jensen (1975) model, terrestrial evaporation is derived from potential evaporation which we computed using the Penman-Monteith equation. Overall, results indicate that, through a simple coupling procedure, the MEP model offers a physically constrained approach to simulate terrestrial evaporation in hydrological models. This approach may offer a tool to better assess climate change impacts on water resources, although the predictive ability of the MEP model under environmental change, may it be natural (e.g. wildfires) or anthropogenic (e.g.
land cover change, climate change), still needs to be assessed. This study focused on the simulation of vertical water fluxes but, to use HGS-MEP for flow simulation and projection, lateral fluxes will need to be considered in further work. Various routing models are available and could be used in conjunction with HGS-MEP to simulate lateral fluxes in a computationally efficient way. Finally, the present study focused on the application of the HGS-MEP model at snow-free sites and the MEP model has undergone little testing in cold regions, with tests limited to the snow-free period (Wang et al., 2017). A MEP
model for snow surfaces (Wang et al., 2014) is available and could also be integrated to hydrological models to allow energy budget modeling throughout the year in northern environments.

**Data availability**

Data are available upon request from the corresponding author. For AmeriFlux data, see Baldocchi (2016), Meyers (2016) and Scott (2016).

**Supplement**

A Supplement related to this article is available.

**Acknowledgements**

This research was supported by NSERC (grants RDC/477125-14 and RGPIN/04892-2015), Hydro-Québec, Ouranos, and Environment and Climate Change Canada. We thank Dr. Jingfeng Wang (Georgia Tech) for his help with the implementation
of the MEP model. Funding for AmeriFlux data resources was provided by the U.S. Department of Energy's Office of Science. We thank the primary investigators of the AmeriFlux sites that made this research possible by making their data available: Russel Scott (US-Wkg), Dennis Baldocchi (US-Ton) and Tilden Meyers (US-WBW). All data used in this paper are listed in the tables and references.

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
