# Peer review of "Using the Maximum Entropy Production approach to integrate energy budget modeling in a hydrological model"

_Hydrology and Earth System Sciences, 2018_

## Referee Comment (RC1) · Erwin Zehe (Referee) · 9 Apr 2019

Summary: The authors use the MEP constrained energy balance model derived Wang and Bras (2011) to simulate evaporation with the hydrological model "Hydro-GeoSphere". More specifically they couple the MEP energy balance model with the model "HydroGeoSphere" (HGS-MEP), and evaluate their approach within a long term uncalibrated simulation against energy flux data and soil moisture data of three distinctly different sites. Moreover, the authors compare their HGS-MEP model to the HydroGeoSphere standard using the Penman Monteith approach (HGS-PM) as a null model.

[Figure]

Evaluation: I very much enjoyed the reading of this study as I generally like the idea of using thermodynamic optimality for constraining the land-surface energy balance. I am also in favor of the proposed evaluation strategy. The scientific significance of the study is in principle high, as evaporation is certainly one of the most important fluxes when it comes to change. Moreover, the study is nicely written, well-structured, based on sound data and nicely illustrated. So I would definitely like to see it published in the ESD/HESS SI. Nevertheless, there are several important issues that need to be clarified in a round of major revisions before the study might become acceptable.

Major points:

M 1): From the presentation of the underlying theory it becomes neither clear how entropy production is defined in the model nor how it has been optimized. While I acknowledge that the study relies on an already published model of Wang and Bras (2011), it is important to share this with the readers. There are several fluxes which produce entropy in the soil-atmosphere vegetation system, while they deplete their driving gradients. The sensible heat flux, depleting the near surface gradient in air temperature, the evapo-transpiration flux depleting the gradient in partial water vapor pressure, and also the soil water flow depleting gradients in soil water potentials (e.g. Zehe et al. 2013). To which entropy production term is the model referring to, or is it referring to all?

M 2): The second major point closely relates to the first one. The proposed transpiration model is driven by dependent variables, particularly the relative humidity and the air temperature from the eddy covariance data are not independent form ET and H. I would expect that an optimization of these fluxes with respect to entropy production needs to account for the feedback of these fluxes on these driving gradients, by defining entropy production as flux times the driving potential difference divided by the absolute temperature. As this is not the case here, I wonder about the definition of entropy production (see M1).

M3) Last but not least the long wave upward flux is a function of the surface temperature and the emissivity in the thermal infrared. By using Rn as driver the authors constrain the amount of energy which is available for ET+H+G. This is a substantial constraint for the entropy production as well.

M4): The proposed results underpin very much that the HGS-MEP perform superior. But does it perform acceptable? The latter requires definition of a model acceptance threshold a priory., e.g. of NSE > x. At US-TON the soil water content and ET are underestimated by -5%, -11%. So where did the water go? The authors evaluate their model using daily mean values. I would be interested in seeing the model performance at the diurnal scale.

Minor points:

Line 60: I very much agree that hydrological model applications are largely insensitive to the choice of the ET model. But is this really a surprise? We calibrate the model to reproduce discharge – so do they have an alternative?

The NSE and the RMSE are not independent, so the authors might consider to skip one of the metrics? Page 2 line 45: PM is also constraint by Rn.

Eq. 3: I wonder why thermal inertia of liquid water is weighted by soil water content, thermal inertia of the solid phase is not weighted by the volume fraction of the solid phase.

From a soil physical standpoint field capacity is a scale dependent, the average potential value at which a probe stops gravity driven seepage depends on the height of the probe.

Eq. 15 and 17. I wonder about the definition of Ec.

Eq. 18: Are the theta_e1 and theta_e2 calibrated, if so this is a substantial constraint to entropy production?

Eq. 21:I wonder whether this relation is only valid for neutral conditions?

Figure 6: The deviations between the model and the observed soil water content value appear a little too large for an NSE of 0.61. Please double check.

Best regards,

Erwin Zehe

References: Wang, J. and Bras, R. L.: A model of evapotranspiration based on the theory of maximum entropy production, Water Resour. Res., 47, W03 521, https://doi.org/10.1029/2010WR009392, http://dx.doi.org/10.1029/2010WR009392, 2011. Zehe, E., Ehret, U., Blume, T., Kleidon, A., Scherer, U., and Westhoff, M.: A thermodynamic approach to link self-organization, preferential flow and rainfall-runoff behaviour, Hydrology And Earth System Sciences, 17, 4297-4322, 10.5194/hess-17-4297-2013, 2013.

---

## Referee Comment (RC2) · Axel Kleidon (Referee) · 9 Apr 2019

This paper describes the application of a information-theory based Maximum Entropy Production (MEP) approach to partition turbulent fluxes into sensible and latent heat. The authors use three sites in the US to test this approach and compare it to one that is based on the Penman-Monteith (PM) equation. They showed that the MEP approach appears to work better than PM. I think this is a useful paper that is well written and carefully explained, but at present a little thin on insight. I think that some points need to be clarified, some additional evaluations would help to better interpret the results, and some disadvantages of the methodology need to be discussed before it may be

suitable for publication.

Major comments:

1. Information-based MEP approach: Despite the success in applying the MEP approach that was developed by Jingfeng Wang and that is shown in this manuscript, I have some reservations about the approach. First, by using six measurements, it seems to me that this is already quite a bit of information for the partitioning of sensible and latent heat and is probably already overconstrained. You use net radiation (minus ground heat flux), this already sets the magnitude of the turbulent fluxes, and then it is only a question about partitioning these into sensible and latent heat. Also, the variables are not independent from each other. Net radiation, for instance, combines net solar radiation with downwelling longwave radiation and thermal emission, with the latter being strongly correlated with temperature. So these input fields do not contain independent information. This aspect, however, is nowhere mentioned, discussed, or evaluated.

In addition I feel uneasy about this approach because it is not process-based. So would this approach also be able to predict the right sensitivity to, say, global warming, land cover change, or vegetation-caused phenology changes? It seems to me that with natural vegetation, it may have adapted so well to its environment that one does not see a sign of vegetation, but this may change with human-caused land cover change. So I am doubtful whether this approach can represent such sensitivities, because it is not really based on mechanisms. Because of this absence of mechanisms, I would also not refer to the approach as parsimonious.

I do not expect the authors to solve these issues, but at the minimum, I would expect the authors to discuss these thoroughly and evaluate potential impacts. It would need some critical evaluation of this approach and point out some further needs to evaluate, especially when advocating a non process-based approach.

2. Additional analyses: At the moment, I feel that there is relatively little done in terms

of analysing the conditions when one approach works better or worse than the other. What would help in this direction is to analyse the time periods when soil water or atmospheric demand are the primary limitations to ET. I think this would be easy to do and useful.

Also, I noticed in Fig. 4 that at the US-Ton site, evaporation seems to be consistently underestimated. I could imagine that this has to do with the relatively shallow rooting depths that have been assumed in both modelling approaches. The Tonzi site is in a mediterranean climate, and vegetation there is well known to have deep roots. The model uses rather shallow rooting depths of 1m or less, and such a depth could be too shallow. Also, in the model formulation of water limitation, it weighs root uptake with some sort of cubic decay function. This is not really how roots work. When water is available in a soil layer, it is being taken up if roots are there, and it seems this is fairly independent of biomass. So this formulation may also result in the low evaporation bias during the dry season.

So I think it would be instructive to include a sensitivity analysis to evaluate if both approaches can be improved by better rooting depth parameterisations.

Minor comments:

General:

Why do you use the Penman-Monteith equation as a reference? Milly and Dunne (2016) have, for instance, shown that it can lead to some systematic biases in sensitivity. Have you checked the Priestley-Taylor approach as well that presumably works better?

What is the uncertainty related to the lack of energy balance closure of the eddy flux data?

How do the fluxes look like when evaluated at the time scale of the diurnal cycle? At the moment, only daily means are being evaluated, but the observations should be

available at a higher temporal resolution. So why not look at and evaluate the simulation of the diurnal cycle as well?

Specific:

p4, lines 29-30. How are $C_1$ and $C_2$ "universal" constants? Also, why does the von Karman constant appear in the expressions? I thought the information-based approach does not rely on semi-empirical parameterizations of turbulent fluxes. Please clarify.

p5 Eq. 8. How does this equation for sigma relate to more common expressions in micrometeorology, such as the equilibrium Bowen ratio?

p5, line 32. Why is water uptake weighted by the vertical root distribution? There is quite some evidence for roots being able to take up substantial amounts of soil moisture even at low root biomass concentrations (see e.g., Nepstad et al. (1994) Nature).

p8 lines 10-15. Why did you not use the radiative surface temperature as the skin temperature that can be inferred from the longwave upwelling flux? It seems to me that the radiative temperature would be a more adequate representation of skin temperature.

Milly, P.C.D. and Dunne, K.A. (2016) Potential evaporation and continental drying. Nature Climate Change, 6, 946–949.

---

## Editor Comment (EC1) · Hubert H.G. Savenije (Editor) · 6 May 2019

Dear Authors,

I think you adequately replied to the reviewers' comments. I suggest you submit a fully revised paper which I shall then send again to the two reviewers for their assessment. Please make clear which changes you have made to the original text and motivate these in a separate letter.

Sincerely, Hubert Savenije
* * *
636, 2019.

---

## Author Comment (AC2) · 6 May 2019

Major comments: 1. Information-based MEP approach: Despite the success in applying the MEP approach that was developed by Jingfeng Wang and that is shown in this manuscript, I have some reservations about the approach. First, by using six measurements, it seems to me that this is already quite a bit of information for the partitioning of sensible and latent heat and is probably already overconstrained. You use net radiation (minus ground heat flux), this already sets the magnitude of the turbulent fluxes, and then it is only a question about partitioning these into sensible and latent heat. Also, the variables are not independent from each other. Net radiation, for instance, combines

net solar radiation with downwelling longwave radiation and thermal emission, with the latter being strongly correlated with temperature. So these input fields do not contain independent information. This aspect, however, is nowhere mentioned, discussed, or evaluated.

In addition I feel uneasy about this approach because it is not process-based. So would this approach also be able to predict the right sensitivity to, say, global warming, land cover change, or vegetation-caused phenology changes? It seems to me that with natural vegetation, it may have adapted so well to its environment that one does not see a sign of vegetation, but this may change with human-caused land cover change. So I am doubtful whether this approach can represent such sensitivities, because it is not really based on mechanisms. Because of this absence of mechanisms, I would also not refer to the approach as parsimonious. I do not expect the authors to solve these issues, but at the minimum, I would expect the authors to discuss these thoroughly and evaluate potential impacts. It would need some critical evaluation of this approach and point out some further needs to evaluate, especially when advocating a non process-based approach.

RESPONSE: Before addressing the predictive ability of MEP under change, we believe the first step was to demonstrate the interest of this approach under relatively stationary conditions. Indeed, very little work has been done on the evaluation of the MEP model. The present study is indeed among the first to test the model under different climates and for multiple years, as Wang et al. (2011) had only presented short-term proof of concepts, mainly in a semi-arid climate. Issues raised by the reviewer are indeed important and we will address them in the discussion. We will raise the issue of non-independence in section 5.2 of the discussion to recognize this limitation to the MEP model. Moreover, we will modify the text and remove references to the parsimony of the model. Regarding the sensitivity of the model to change, we will raise this point in the conclusion and stress the need to address this important question in further work.

2. Additional analyses: At the moment, I feel that there is relatively little done in term

of analysing the conditions when one approach works better or worse than the other. What would help in this direction is to analyse the time periods when soil water or atmospheric demand are the primary limitations to ET. I think this would be easy to do and useful.

RESPONSE: We will perform additional analyses. First, we will add a section describing the performance of the models at the diurnal scale, as per Reviewer 1 suggestion. Second, as suggested here, we will compare the performance of the model under energy vs. water-limiting conditions. To do so, we will compute the monthly aridity index (ratio between precipitation and potential evapotranspiration) and periods with a monthly index > 1 will be considered energy-limited and periods with a monthly index < 1 will be considered water-limited. We will finally compute performance metrics to compare the performance of the HGS-MEP and HGS-PM models for these two periods.

Also, I noticed in Fig. 4 that at the US-Ton site, evaporation seems to be consistently underestimated. I could imagine that this has to do with the relatively shallow rooting depths that have been assumed in both modelling approaches. The Tonzi site is in a mediterranean climate, and vegetation there is well known to have deep roots. The model uses rather shallow rooting depths of 1m or less, and such a depth could be too shallow. Also, in the model formulation of water limitation, it weighs root uptake with some sort of cubic decay function. This is not really how roots work. When water is available in a soil layer, it is being taken up if roots are there, and it seems this is fairly independent of biomass. So this formulation may also result in the low evaporation bias during the dry season. So I think it would be instructive to include a sensitivity analysis to evaluate if both approaches can be improved by better rooting depth parameterisations.

RESPONSE: Regarding the rooting depth at US-Ton, we had performed tests and modelled terrestrial evaporation in stand-alone MEP mode, using soil water content observations as an input variable (Figure R1). Soil water content observations nearest

to the surface were used as input to the MEP-Ev model (z = 0 cm at US-Ton) and observations in the middle soil layer were used as input to the MEP-Tr model (z = 20 cm at US-Ton). While the HGS-MEP simulates soil moisture very well at a depth of 20 cm (Figure 5, p.14), it tends to underestimate soil moisture close to the surface, thus overestimating the water stress and limiting near surface water uptake and at the same time, transpiration. When using soil moisture observation, we avoid this situation and instead have soil conditions with greater water availability. As shown in Figure R1 below, access to a greater water supply did not improve the simulation of evapotranspiration. Instead of underestimation, we now face a large overestimation of terrestrial evaporation in the second half of the year. These results suggest that a greater rooting depth that would allow vegetation to tap deep water resources is not likely to improve the simulation of terrestrial evaporation at US-Ton. Instead, uncertainty relative to the definition of water stress points (wilting point and field capacity), as discussed on p.17 (line 4), may explain the underestimation of the terrestrial evaporation at this site.

Regarding the weighting of water uptake based on a cubic decay function, it is very common in hydrological or land surface models to weight vegetation water uptake based on the vertical root density (see for instance equation 4, Feddes et al., 2001). Using this approach, water uptake in a given soil layer depends on the root fraction in this particular layer. The parameterization of root water uptake is the subject of active research (see Clark et al., 2015 for a review) and while important, it is not the main focus of the present study.

Minor comments:

General: Why do you use the Penman-Monteith equation as a reference? Milly and Dunne (2016) have, for instance, shown that it can lead to some systematic biases in sensitivity. Have you checked the Priestley-Taylor approach as well that presumably works better?

RESPONSE: The main objective of the study was to assess the predictive ability of the

MEP model and various benchmarks could have been used. We chose the Penman-Monteith model as it is a theoretically-sound model of terrestrial evaporation. In our experience, the MEP model has been met with a certain reluctance given its roots in information theory, thus our choice of the process-based Penman-Monteith model as a benchmark. We will add a few sentences in the discussion to point out the systematic bias observed with the Penman-Monteith model, as shown by Milly and Dunne (2016).

What is the uncertainty related to the lack of energy balance closure of the eddy flux data?

RESPONSE: We did not quantify the uncertainty associated with the lack of energy balance closure for the eddy flux data. We will add text to the discussion to raise this additional source of uncertainty. However, since we are mostly interested in a comparison between models, we can expect their performance to be similarly impacted by the lack of closure of the energy balance.

How do the fluxes look like when evaluated at the time scale of the diurnal cycle? At the moment, only daily means are being evaluated, but the observations should be available at a higher temporal resolution. So why not look at and evaluate the simulation of the diurnal cycle as well?

RESPONSE: As suggested by Reviewer 1, we will add an analysis of the performance of the models at the diurnal scale.   Specific:

p4, lines 29-30. How are C1 and C2 "universal" constants? Also, why does the von Karman constant appear in the expressions? I thought the information-based approach does not rely on semi-empirical parameterizations of turbulent fluxes. Please clarify.

RESPONSE: We will remove the term "universal" as it can be confusing. As for the von Kármán constant, it is involved in the calculation of the apparent thermal inertia of air given the latter is derived from an extremum solution of the Monin-Obukhov similarity equations.

[Figure]

p5 Eq. 8. How does this equation for sigma relate to more common expressions in micrometeorology, such as the equilibrium Bowen ratio?

RESPONSE: The Bowen ratio, as predicted by the MEP model, agrees with the ratio derived with the Priestley-Taylor model, as demonstrated by Wang et al. (2011; Figure 1).

p5, line 32. Why is water uptake weighted by the vertical root distribution? There is quite some evidence for roots being able to take up substantial amounts of soil moisture even at low root biomass concentrations (see e.g., Nepstad et al. (1994) Nature).

RESPONSE: As stated above, it is very common in hydrological or land surface models to determine the depth of vegetation water uptake based on the vertical root density. Improving the parameterization of root water uptake was not the focus of the present study.

p8 lines 10-15. Why did you not use the radiative surface temperature as the skin temperature that can be inferred from the longwave upwelling flux? It seems to me that the radiative temperature would be a more adequate representation of skin temperature.

RESPONSE: The longwave upwelling flux is measured above the canopy at US-Ton (z = 23 m) and US-WBW (z= 36.9 m), which we do not believe would offer a good proxy of the skin temperature when considering the soil surface.

REFERENCES: Clark et al. (2015) Improving the representation of hydrologic processes in EarthSystem Models, Water Resources Research, 51:5929-5956. Feddes et al. (2001) Modeling Root Water Uptake in Hydrological and Climate Models. Bulletin of the American Meteorological Society, 82(12):2797-2809.

———————————————————

[Figure]

Figure R1. Comparison of observed evapotranspiration and evapotranspiration simulated by the HGS-MEP model and by the MEP-ET model using soil water content (SWC) observations at US-Ton

**Fig. 1.**

---

## Author Response (AR1)

Below is our answer to comments from reviewer #1 and #2. In red is our answer as detailed in our Author Comments. In purple is the modification performed in the manuscript. All line numbers refer to the manuscript in track changes mode, as found in the present document below our answer to reviewers.

REVIEWER #1

Summary: The authors use the MEP constrained energy balance model derived Wang and Bras (2011) to simulate evaporation with the hydrological model "HydroGeoSphere". More specifically they couple the MEP energy balance model with the model "HydroGeoSphere" (HGS-MEP), and evaluate their approach within a long term uncalibrated simulation against energy flux data and soil moisture data of three distinctly different sites. Moreover, the authors compare their HGS-MEP model to the HydroGeoSphere standard using the Penman Monteith approach (HGS-PM) as a null model.

Evaluation: I very much enjoyed the reading of this study as I generally like the idea of using thermodynamic optimality for constraining the land-surface energy balance. I am also in favor of the proposed evaluation strategy. The scientific significance of the study is in principle high, as evaporation is certainly one of the most important fluxes when it comes to change. Moreover, the study is nicely written, well-structured, based on sound data and nicely illustrated. So I would definitely like to see it published in the ESD/HESS SI. Nevertheless, there are several important issues that need to be clarified in a round of major revisions before the study might become acceptable.

Major points:

M1): From the presentation of the underlying theory it becomes neither clear how entropy production is defined in the model nor how it has been optimized. While I acknowledge that the study relies on an already published model of Wang and Bras (2011), it is important to share this with the readers. There are several fluxes which produce entropy in the soil-atmosphere vegetation system, while they deplete their driving gradients. The sensible heat flux, depleting the near surface gradient in air temperature, the evapotranspiration flux depleting the gradient in partial water vapor pressure, and also the soil water flow depleting gradients in soil water potentials (e.g. Zehe et al. 2013). To which entropy production term is the model referring to, or is it referring to all?

RESPONSE: The entropy production term refers to the surface heat fluxes. We will better emphasize this point in the description of the model (section 3.1).

MODIFICATION: We modified section 3.1 to specify that the MEP model "*uses this theory (MaxEnt) to model the land surface energy balance*" (p. 4, line 20).

M2): The second major point closely relates to the first one. The proposed transpiration model is driven by dependent variables, particularly the relative humidity and the air temperature from the eddy covariance data are not independent form ET and H. I would expect that an optimization of these fluxes with respect to entropy production needs to account for the feedback of these fluxes on these driving gradients, by defining entropy production as flux times the driving potential difference divided by the absolute temperature. As this is not the case here, I wonder about the definition of entropy production (see M1).

RESPONSE: The definition of entropy as a flux times the driving potential difference divided by the absolute temperature refers to *thermodynamic* entropy. In Wang's MEP model use in the present study, entropy refers to Shannon entropy (see line 34, p.2), that is a quantitative measure of information. The MEP model is derived from the principle of maximum entropy (MaxEnt) developed by Jaynes (1957) as a general method to assign probability distribution in statistical mechanics (see line 34 p.2). This point is re-emphasized in the Methodology (see line 12, p.4). To avoid any confusion between these two definitions of entropy, we will clearly state in the Methodology that entropy does not refer to thermodynamic entropy.

MODIFICATION: We added the following text to section 3.1 to clearly state that the MEP model does not rely on the concept of thermodynamic entropy:
"In the MEP model, entropy refers to Shannon entropy, which is the expected value of information (Shannon, 1948), which is not related to thermodynamic entropy expressed as the ratio of flux to temperature. Indeed, the MEP model is derived from the principle of maximum entropy (MaxEnt) developed by Jaynes (1957) as a general inference tool to assign probability distributions in statistical mechanics. Wang and Bras (2009) used this statistical approach to develop the MEP model of land surface fluxes." (p. 4, lines 17-21).

M3) Last but not least the long wave upward flux is a function of the surface temperature and the emissivity in the thermal infrared. By using Rn as driver the authors constrain the amount of energy which is available for ET+H+G. This is a substantial constraint for the entropy production as well.

RESPONSE: Indeed, the closure of the energy budget is an important constraint to entropy production, but conservation of energy is a fundamental principle controlling surface heat fluxes. Other models of terrestrial evaporation are built around this constraint. For example, in the Penman model, evapotranspiration is constrained by the available energy (i.e. net radiation). The MEP model is based on the same fundamental principle and uses it to develop a predictive tool of the surface heat fluxes. We will add text to the methodology to reinforce the fact that the two models compared in the study, the MEP and Penman-Monteith models, are built around this constraint.

MODIFICATION: We added the following text to section 3.4.2 to describe the Penman-Monteith equation as *"a physically based model where, similar to the MEP model, the predicted terrestrial evaporation is constrained by available energy at the surface"* (p.9, lines 15-16).

M4): The proposed results underpin very much that the HGS-MEP perform superior. But does it perform acceptable? The latter requires definition of a model acceptance threshold a priory.,e.g. of NSE > x. At US-TON the soil water content and ET are underestimated by -5%, -11%. So where did the water go? The authors evaluate their model using daily mean values. I would be interested in seeing the model performance at the diurnal scale.

RESPONSE: We agree that this would be a very interesting addition to the manuscript. We will add a section describing the performance of the models at the diurnal scale, as depicted in figure R1. HGS-MEP performed well in describing the diurnal pattern of variation in terrestrial evaporation, although we observed an overestimation of maximum values at mid-day. In addition to the figure, we will also add a table with performance metrics to quantify the ability of the models to capture the diurnal variation in terrestrial evaporation.

MODIFICATION: We added a paragraph to discuss the performance of HGS-MEP at US-Ton (p.19, lines 17-31) and its relation to the rooting depth. Moreover, we added results regarding the performance of models at a diurnal scale. We added Figure 5 which shows the observed and modeled hourly average terrestrial evaporation at the three sites. We also added Table 6 with performance metrics computed at a half-hourly time scale. We present these results for each site (p.11 line 43, p.5 line 33, p.17 line 17). We also added a paragraph regarding these results in the discussion (p. 19 lines 31-40).

Minor points:

Line 60: I very much agree that hydrological model applications are largely insensitive to the choice of the ET model. But is this really a surprise? We calibrate the model to reproduce discharge – so do they have an alternative?

RESPONSE: We agree with this point. At line 33 (p.1), we wanted to highlight the fact that hydrologic simulations are generally not much sensitive to the choice of ET model, to better stress out its large influence when performing hydrological projections (line 35). The alternative is to attempt to model hydrologic fluxes without relying on automatic calibration but instead using a priori estimation of parameters from soil and vegetation data (Wagener, 2007). This is in fact the approach implemented by the present study, as no automatic calibration was performed. We stress this point a few times in the discussion (line 2 and 9, p.17, line 44 p.18). We will add one of two sentences in the discussion to stress the fact that a priori estimation of parameter shows promising results in terms of the predictive ability of the model, thus offering an alternative to calibration.

MODIFICATION: We added the following text to section 5.1.1: "The predictive ability of the MEP model is particularly noteworthy given that we did not rely on calibration and instead used a priori estimation of parameters describing the soil and vegetation. The MEP model thus offers a promising alternative to model hydrologic fluxes without relying on calibration (Wagener, 2007)" (p.19, lines 41-44).

The NSE and the RMSE are not independent, so the authors might consider to skip one of the metrics?

RESPONSE: Yes, we know that these metrics are not independent, but given that they are commonly reported, we would like to keep both. We will add a sentence in the Methodology to recognize the fact that the metrics are not independent.

MODIFICATION: We added the following text to section 3.6: "The RMSE and NSE are not independent metrics of performance given the NSE is only a normalised measure of the mean square error. Still, we chose to report the two metrics

as both are commonly reported and provide an assessment of performance in absolute (RMSE) and relative (NSE) terms." (p.10, lines 32-34).

Page 2 line 45: PM is also constraint by Rn.

RESPONSE: Yes, we will add a sentence to highlight this fact.

MODIFICATION: In line with modifications made in response to comment M3, we indicated that the Penman-Monteith model is "a physically based model where, similar to the MEP model, the predicted terrestrial evaporation is constrained by available energy at the surface" (p.9, lines 15-16).

Eq. 3: I wonder why thermal inertia of liquid water is weighted by soil water content, thermal inertia of the solid phase is not weighted by the volume fraction of the solid phase.

RESPONSE: The equation to compute soil thermal inertia is empirically derived and was directly taken from Huang and Wang (2016), as cited p.5 (line 20). We will add a sentence to highlight the empirical nature of the equation. Still, as mentioned in section 3.3 and 3.4, the MEP model is not very sensitive to changes in soil thermal inertia. This parameter was in fact set as a constant equal to the dry soil thermal inertia. Accordingly, any modification to the formula used to compute soil thermal inertia would have little impact on the results of our simulations.

MODIFICATION: We added that soil thermal inertia is calculated "with the empirically-derived equation from Huang and Wang (2016)" (p.4, line 31).

From a soil physical standpoint field capacity is a scale dependent, the average potential value at which a probe stops gravity driven seepage depends on the height of the probe.

RESPONSE: Field capacity has been defined based on soil matric potential (-0.033 MPa). This definition is common in hydrology (e.g. Dingman, 2002). We defined field capacity accordingly, as detailed on p.8, line 1.

Eq. 15 and 17. I wonder about the definition of Ec.

RESPONSE: Ec corresponds to the wet canopy evaporation. Since, as stated on p.9 (line 12), interception was not considered in the present study, $Ec$ was set to zero, as specified on p.9 line 13.

Eq. 18: Are the theta_e1 and theta_e2 calibrated, if so this is a substantial constraint to entropy production?

RESPONSE: No calibration was performed for these parameters. We stress this fact a few times in the manuscript and modified the manuscript to further highlight it as suggested by Reviewer 1 (p.19, line 41). Moreover, the definition of theta_e1 and theta_e2 is clearly stated on p. 7 line 8.

Eq. 21: I wonder whether this relation is only valid for neutral conditions?

RESPONSE: Indeed, equation 21 assumes a neutral atmosphere. It has been used with success in other studies modelling terrestrial evaporation (e.g., Ershadi et al., 2014). Still, we will add a sentence to highlight the fact that the estimation of aerodynamic resistance is a source of uncertainty in the Penman-Monteith model.

MODIFICATION: We added the following sentence to section 3.4.2: "Equation 21 was derived for neutral atmospheric conditions but has also been successfully used to model terrestrial evaporation over a wide range of conditions (Ershadi et al., 2014)" (p.9 line 26).

Figure 6: The deviations between the model and the observed soil water content value appear a little too large for an NSE of 0.61. Please double check.

RESPONSE: We double-checked this number and the NSE is 0.61 when comparing observed and simulated daily soil water content at US-WBW. Figure R1 shows that the performance of HGS-MEP using a scatterplot, which shows moderate performance of HGS-MEP in line with a NSE value of 0.61.

[Figure]

Figure R1. Observed and simulated daily soil water content at US-WBW.

Best regards,
Erwin Zehe

References: Wang, J. and Bras, R. L.: A model of evapotranspiration based on the theory of maximum entropy production, Water Resour. Res., 47, W03 521,https://doi.org/10.1029/2010WR009392,http://dx.doi.org/10.1029/2010WR009392,2011. Zehe, E., Ehret, U., Blume, T., Kleidon, A., Scherer, U., and Westhoff, M.: A thermodynamic approach to link self-organization, preferential flow and rainfall-runoff behaviour, Hydrology And Earth System Sciences, 17, 4297-4322, 10.5194/hess-17-4297-2013, 2013.

OTHER MODIFICATIONS: Given recent work available on MEP, we added two sentences in to introduction to put our work in this context (p.3 lines 3-7). We also performed a few minor edits (word change, adding a precision, etc) which are all shown in the track changes document.

RESPONSE: We will perform additional analyses. First, we will add a section describing the performance of the models at the diurnal scale, as per Reviewer 1 suggestion. Second, as suggested here, we will compare the performance of the model under energy vs. water-limiting conditions. To do so, we will compute the monthly aridity index (ratio between precipitation and potential evapotranspiration) and periods with a monthly index > 1 will be considered energy-limited and periods with a monthly index < 1 will be considered water-limited. We will finally compute performance metrics to compare the performance of the HGS-MEP and HGS-PM models for these two periods.

MODIFICATION: First, we added results regarding the performance of models at a diurnal scale. We added Figure 5 which shows the observed and modeled hourly average terrestrial evaporation at the three sites. We also added Table 6 with performance metrics computed at a half-hourly time scale. We present these results for each site (p.11 line 43, p.5 line 33, p.17 line 17). We also added a paragraph regarding these results in the discussion (p.19 lines 31-40).

Second, we also added results regarding the performance of models during wet and dry periods. As detailed on p.11 (lines 1-8), we computed the monthly aridity index, that is the ratio between precipitation and PET. Using monthly values of the aridity index, we then calculated performance metrics for dry periods (P/PET < 0.4 at US-Wkg and P/PET < 1 at US-Ton and US-WBW) and wet periods (P/PET ≥ 0.4 at US-Wkg and P/PET ≥ 1 at US-Ton and US-WBW). We added Table 5 with performance metrics for wet and dry periods and we present these results for each site (p.11 line 38, p.15 line 28, p.17 line 12). We also added a paragraph regarding these results in the discussion (p.18 lines 8-14).

Also, I noticed in Fig. 4 that at the US-Ton site, evaporation seems to be consistently underestimated. I could imagine that this has to do with the relatively shallow rooting depths that have been assumed in both modelling approaches. The Tonzi site is in a mediterranean climate, and vegetation there is well known to have deep roots. The model uses rather shallow rooting depths of 1m or less, and such a depth could be too shallow. Also, in the model formulation of water limitation, it weighs root uptake with some sort of cubic decay function. This is not really how roots work. When water is available in a soil layer, it is being taken up if roots are there, and it seems this is fairly independent of biomass. So this formulation may also result in the low evaporation bias during the dry season. So I think it would be instructive to include a sensitivity analysis to evaluate if both approaches can be improved by better rooting depth parameterisations.

RESPONSE: Regarding the rooting depth at US-Ton, we had performed tests and modelled terrestrial evaporation in stand-alone MEP mode, using soil water content observations as an input variable (Figure R2). Soil water content observations nearest to the surface were used as input to the MEP-Ev model (z = 0 cm at US-Ton) and observations in the middle soil layer were used as input to the MEP-Tr model (z = 20 cm at US-Ton). While the HGS-MEP simulates soil moisture very well at a depth of 20 cm (Figure 5, p.14), it tends to underestimate soil moisture close to the surface, thus overestimating the water stress and limiting near surface water uptake and at the same time, transpiration. When using soil moisture observation, we avoid this situation and instead have soil conditions with greater water availability. As shown in Figure R1 below, access to a greater water supply did not improve the simulation of evapotranspiration. Instead of underestimation, we now face a large overestimation of terrestrial evaporation in the second half of the year. These results suggest that a greater rooting depth that would allow vegetation to tap deep water resources is not likely to improve the simulation of terrestrial evaporation at US-Ton. Instead, uncertainty relative to the definition of water stress points (wilting point and field capacity), as discussed on p.17 (line 4), may explain the underestimation of the terrestrial evaporation at this site.

Regarding the weighting of water uptake based on a cubic decay function, it is very common in hydrological or land surface models to weight vegetation water uptake based on the vertical root density (see for instance equation 4, Feddes et al., 2001). Using this approach, water uptake in a given soil layer depends on the root fraction in this particular layer. The parameterization of root water uptake is the subject of active research (see Clark et al., 2015 for a review) and while important, it is not the main focus of the present study.

[Figure]

Figure R2. Comparison of observed terrestrial evaporation and terrestrial evaporation simulated by the HGS-MEP model and by the MET-E model using soil water content (SWC) observations at US-Ton.

MODIFICATION: We added the following paragraph to provide potential explanations of the negative bias observed in modelled terrestrial evaporation at US-Ton: "At the Mediterranean site US-Ton, terrestrial evaporation was underestimated (PBIAS = -14 %, Table 4), particularly during the second half of the year as reduced water supply leads to a reduction in terrestrial evaporation (Figure 5). While the HGS-MEP model simulates soil moisture very well at a depth of 20 cm (Figure X), it tended to underestimate soil moisture close to the surface (Figure S2), where the largest proportion of root is found according to the vertical root distribution defined by HGS (cubic decay distribution between the surface and the maximum

root depth). Given that the water stress factor ($\eta_s$) is computed from the weighted average soil water content over the root zone (equation 12), an underestimation of soil moisture near the surface translates into an overestimation of the reduction in transpiration resulting from water stress. We investigated if this issue could be due to misdefinition of the maximum rooting depth, as trees under a Mediterranean climate have been found to access water from deep soil layers or groundwater (Miller *et al.*, 2010a). However, we also simulated terrestrial evaporation with the stand-alone MEP model using soil moisture observations, thus avoiding the overestimation of water stress near the surface, and instead found a large overestimation of terrestrial evaporation (Figure S4). These results suggest that increasing the rooting depth to increase access to water resources would likely not improve the simulation of terrestrial evaporation. Instead, uncertainty relative to the definition of the vertical root distribution (as opposed to the maximum rooting depth) or, as previously discussed, with the definition of water stress points (wilting point and field capacity) may explain the challenge of simulating terrestrial evaporation under water-limited conditions at US-Ton." (p.19, lines 15-29).

Minor comments:

General: Why do you use the Penman-Monteith equation as a reference? Milly and Dunne (2016) have, for instance, shown that it can lead to some systematic biases in sensitivity. Have you checked the Priestley-Taylor approach as well that presumably works better?

RESPONSE: The main objective of the study was to assess the predictive ability of the MEP model and various benchmarks could have been used. We chose the Penman-Monteith model as it is a theoretically-sound model of terrestrial evaporation. In our experience, the MEP model has been met with a certain reluctance given its roots in information theory, thus our choice of the process-based Penman-Monteith model as a benchmark. We will add a few sentences in the discussion to point out the systematic bias observed with the Penman-Monteith model, as shown by Milly and Dunne (2016).

MODIFICATION: We added the following sentences to give context on the bias observed for the Penman-Monteith model in other studies: "The predictive ability of the MEP model is particularly noteworthy given that we did not rely on calibration and instead used a priori estimation of parameters describing the soil and vegetation. The MEP model thus offers a promising alternative to model hydrologic fluxes without relying on calibration (Wagener, 2007). We chose the Penman-Monteith as a benchmark against which to compare the MEP model as it is a physically based model that allows for a detailed parameterization of vegetation. However, studies have shown that the Penman-Monteith leads to an underestimation of terrestrial evaporation under a contemporary climate (Ershadi *et al.*, 2014) and an overestimation under climate change (Milly and Dunne, 2016). Other models could have been considered, although Hajji et al. (2018) still demonstrated the superior performance of the MEP model compared to the modified Priestley-Taylor-Jet Propulsion Laboratory (PT-JPL) and the air-relative-humidity-based two-source model (ARTS)" (p.19, lines 41 – p.20 line 3).

What is the uncertainty related to the lack of energy balance closure of the eddy flux data?

RESPONSE: We did not quantify the uncertainty associated with the lack of energy balance closure for the eddy flux data. We will add text to the discussion to raise this additional source of uncertainty. However, since we are mostly interested in a comparison between models, we can expect their performance to be similarly impacted by the lack of closure of the energy balance.

MODIFICATION: We added text to address the issue of energy imbalance when discussing the performance of the model at a half-hourly time scale (p. 19 lines 36-40).

How do the fluxes look like when evaluated at the time scale of the diurnal cycle? At the moment, only daily means are being evaluated, but the observations should be available at a higher temporal resolution. So why not look at and evaluate the simulation of the diurnal cycle as well?

RESPONSE: As suggested by Reviewer 1, we will add an analysis of the performance of the models at the diurnal scale.

MODIFICATION: We added results regarding the performance of models at a diurnal scale. We added Figure 5, which shows the observed and modeled hourly average terrestrial evaporation at the three sites. We also added Table 6 with performance metrics computed at a half-hourly time scale. We present these results for each site (p.11 line 43, p.5 line 33, p.17 line 17). We also added a paragraph regarding these results in the discussion (p. 19 lines 31-40).

Specific:

p4, lines 29-30. How are C1 and C2 "universal" constants? Also, why does the von Karman constant appear in the expressions? I thought the information-based approach does not rely on semi-empirical parameterizations of turbulent fluxes. Please clarify.

RESPONSE: We will remove the term "universal" as it can be confusing. As for the von Kármán constant, it is involved in the calculation of the apparent thermal inertia of air given the latter is derived from an extremum solution of the Monin-Obukhov similarity equations.

MODIFICATION: We removed the term "universal" on p.5 (line 2).

p5 Eq. 8. How does this equation for sigma relate to more common expressions in micrometeorology, such as the equilibrium Bowen ratio?

RESPONSE: The Bowen ratio, as predicted by the MEP model, agrees with the ratio derived with the Priestley-Taylor model, as demonstrated by Wang et al. (2011; Figure 1).

p5, line 32. Why is water uptake weighted by the vertical root distribution? There is quite some evidence for roots being able to take up substantial amounts of soil moisture even at low root biomass concentrations (see e.g., Nepstad et al. (1994) Nature).

RESPONSE: As stated above, it is very common in hydrological or land surface models to determine the depth of vegetation water uptake based on the vertical root density. Improving the parameterization of root water uptake was not the focus of the present study.

p8 lines 10-15. Why did you not use the radiative surface temperature as the skin temperature that can be inferred from the longwave upwelling flux? It seems to me that the radiative temperature would be a more adequate representation of skin temperature.

RESPONSE: The longwave upwelling flux is measured above the canopy at US-Ton (z = 23 m) and US-WBW (z= 36.9 m), which we do not believe would offer a good proxy of the skin temperature when considering the soil surface.

OTHER MODIFICATIONS: Given recent work available on MEP, we added two sentences in to introduction to put our work in this context (p.3 lines 3-7). We also performed a few minor edits (word change, adding a precision, etc) which are all shown in the track changes document.

[revised manuscript text omitted]

---

## Author Response (AR2)

Below is our answer in red to comments from Reviewer #1. All line numbers refer to the manuscript in track changes mode, as found in the present document below our answer to Reviewer #1.

Suggestions for revision or reasons for rejection (will be published if the paper is accepted for final publication)

The revised manuscript has been considerably improved and the authors clarified most of the points I outlined in my first assessment.

Yet I'd like to stress that thermodynamic entropy and information entropy are not independent. They are well connected in the field of statistical mechanics; in fact one can show that thermodynamic entropy equals Shannon entropy times the Boltzmann constant when relating the Shannon entropy of micro-canonical ensemble with the number of micro states the same represent the energetic macro state of a system (compare for instance Hohnerkamp and Römer, classical theoretical physics). And the fact that the authors claim that entropy shall me maximized makes only sense in the light of the second law of thermodynamics, and not for information entropy itself. The entire discussion relates also to the question where energy is dissipated and entropy is produced, through internal dissipation or through exchange at the boundaries.

RESPONSE: We added the following sentence to the introduction to explicit the link between information and thermodynamic entropy (p. 2, lines 35-37) : Both concepts of entropy are linked: the maximisation of Shannon information entropy can be used to assess the probability of a given state for any kind of system and as such, thermodynamic entropy can be considered a special case of Shannon information theory

Secondly I'd like to stick to the comment that long wave radiation is not an independent variable. In my comment I didn't mean that the authors shall violate energy conservation, but that a free optimization would imply include long wave radiation as a free variable (as function surface temperature etc.).

RESPONSE: Reviewer #2 also raised the issue of non-independence and we believe we have already addressed this issue in our answer to his comment, although this element was not stressed in our answer to Reviewer #1 comment about long wave radiation. As detailed in our answer to Reviewer #2, we had added the following sentence to the discussion (p.21, lines 31-33): 
[revised manuscript text omitted]